# ZeroP: Zero-Shot Quantization via Proxy Data

## Abstract

Zero-shot quantization (ZSQ) is a promising approach for achieving low-bit constraint networks without relying on the original data (OD). However, due to the high cost and privacy concerns associated with OD, it is often scarce, leading to the unsatisfactory performance of ZSQ. Most ZSQ methods rely solely on synthetic data (SD) to mitigate this issue. In this paper, we propose a novel ZSQ framework, named ZeroP, that leverages publicly available data - proxy data (PD) - as a substitute for the OD. We first explore the impact of PD on the performance of current ZSQ methods over 16 different computer vision datasets and introduce a simple and effective PD selection method based on batch-normalization statistics to select the optimal PD. We then apply ZeroP to three state-of-the-art pure-SD (using only SD) methods, achieving 7% to 16% improvements in accuracy for MobileNetV1 on ImageNet-1K in a 4-bit setting. Furthermore, we demonstrate the effectiveness of ZeroP on extensive models and datasets. For example, ZeroP achieves a top-1 accuracy of 72.17% for ResNet-50 on ImageNet-1K in a 4-bit setting, outperforming the SOTA pure-SD method by 3.9%. Overall, our results indicate that ZeroP offers a promising solution for achieving high-performance low-bit networks without relying on OD and opens up new avenues for using publicly available data for data-free tasks.

## 1 Introduction

As Deep Learning (DL) continues to advance, DL-based applications have become prevalent in many fields, including Computer Vision (He et al., 2016; Ren et al., 2015; Chen et al., 2017), Natural Language Processing (Bahdanau et al., 2014; Cho et al., 2014), and other domains (Silver et al., 2016). Quantization (Courbariaux et al., 2015; Jacob et al., 2018) has gained popularity in both academia and industry, aiming to obtain low-bit networks with minimal performance loss compared to the pre-trained full-precision model. However, directly quantizing the full-precision network to a limited-bit architecture often leads to significant accuracy degradation. To maintain the performance of quantized models, one approach is to fine-tune them using the original training data (OD) (Cai et al., 2017; Louizos et al., 2018). Nevertheless, acquiring OD can be difficult due to accessibility, cost, or privacy concerns. For example, obtaining a small number of diagnostic medical images may take years. To address this challenge, zero-shot quantization (ZSQ) (Nagel et al., 2019) has been proposed, which performs quantization without requiring access to the OD.

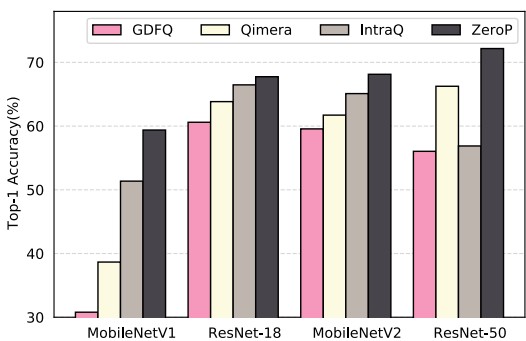

Figure 1: **4-Bit performance comparison on ImageNet-1K.** The top-1 accuracy for 4 architectures. ZeroP significantly outperforms SOTA pure-SD methods, including GDFQ (Xu et al., 2020), Qimera (Choi et al., 2021), and IntraQ (Zhong et al., 2022b), *e.g.*, achieving an 8.02% improvement for MobileNetV1 (Howard et al., 2017) over IntraQ.

However, the absence of access to the OD often leads to significant performance degradation for ZSQ methods (Nagel et al., 2019). To address this issue, two main approaches have been explored in ZSQ. The first approach completely restricts the use of any access data (Guo et al., 2022), rep-

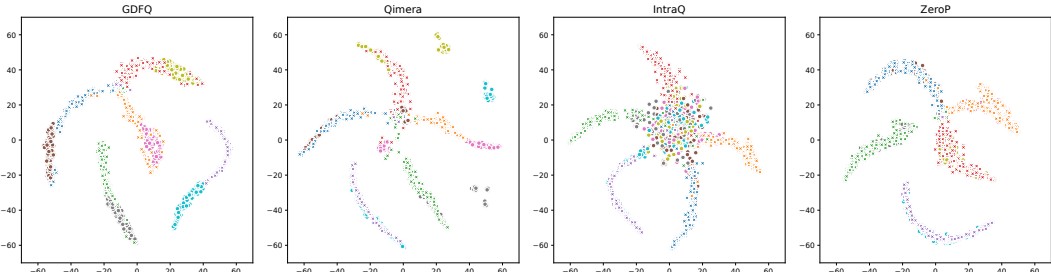

Figure 2: **Visualization of features using t-SNE.** We randomly selected 1000 images (5 classes, 200 images per class) from ImageNet-1K, as well as synthetic images/data (SD) generated by GDFQ, Qimera, IntraQ, and proxy images/data (PD) from COCO used by ZeroP. We used t-SNE (Van der Maaten & Hinton, 2008) to visualize the SD/PD and OD in ImageNet-1K, with 'X' representing OD and 'O' representing SD/PD. The colors indicate different classes, and all class labels were obtained via a pre-trained ResNet-18 (He et al., 2016). Please zoom in for a better visual effect.

resenting the challenging scenario of having no access to OD. In contrast, the other approach (Cai et al., 2020; Xu et al., 2020; Zhang et al., 2021b) aims to mitigate the lack of data by generating synthetic images that incorporate information from the OD. The underlying idea behind synthetic data (SD) is to capture relevant features from the OD and incorporate them into the synthetic samples. The ZeroQ method (Cai et al., 2020) was the first to introduce the concept of using SD in the ZSQ pipeline, and subsequent works (Xu et al., 2020; Zhang et al., 2021b; Zhong et al., 2022b) have explored different aspects of SD generation, such as utilizing generative adversarial networks (Goodfellow et al., 2020) to generate images, diversifying the data distribution (Zhang et al., 2021b), and enhancing intra-class heterogeneity (Zhong et al., 2022b). While these methods attempt to mimic the features of the OD from various angles, they solely rely on SD to achieve performance gains. However, SD may only capture a subset of the variability and complexity present in the OD, as depicted in the first three columns of Fig. 2. Thus, a fundamental question arises:

*Can we obtain or grasp the related information of original data aside from using synthetic data, or must we rely on synthetic data solely?*

In addition to ZSQ, synthetic image techniques have been extensively studied in various related tasks, such as zero-shot knowledge distillation (Nayak et al., 2019; Chen et al., 2019) and black-box model stealing (Orekondy et al., 2019; Barbalau et al., 2020). To augment the diversity of SD, some methods (Addepalli et al., 2020; Orekondy et al., 2019; Sanyal et al., 2022) use proxy data (PD), which is publicly available and easily accessible, to guide the image generation process. However, previous techniques only utilized SD as input data, while PD helped enrich the features of SD. This raises the question of whether PD can be directly used as input data since replacing the OD with PD is not straightforward. One of the immediate challenges is that PD may have a different distribution or be unrelated to OD, leading to reduced performance. Moreover, the selection cost might outweigh its benefits even if the appropriate PD can be identified. In this work, we propose a method to leverage PD to represent OD's information and comprehensively investigate the impact of PD on ZSQ's performance. Our research is intended to bridge the gap between purely synthetic images and directly using PD for quantization fine-tuning.

To do this, we begin by revisiting the core idea behind SD approaches: diversifying sample generation. The main aim of SD methods is to generate data that closely resembles the OD's distribution. However, OD may not always adhere to a single perspective assumption, and capturing its various features can be challenging. For instance, an image cartoonization dataset (Royer et al., 2020) may require area smoothing and sharp edge descriptions instead of class-label information (Deng et al., 2009). Accurately representing the various properties of OD is crucial to address OD deficiency and to improve performance. Hence, understanding whether PD can represent the abundant latent features of OD is essential. To achieve this, we first construct a straightforward ZSQ baseline that employs 16 distinct PDs as direct inputs for quantization fine-tuning rather than PD to guide SD generation. This simple modification results in a significant performance improvement. Next, we devise a simple and computationally feasible method to address the second challenge — how to rapidly and inexpensively choose an effective PD. Specifically, we calculate a scalar distance using

the batch normalization statistic (Ioffe & Szegedy, 2015) (BNS) of the full-precision model for all PDs and select the final PD based on the PD's distance rank. We also conduct a series of experiments to demonstrate that the simple BNS distance can provide informative signals for predicting the efficacy of a PD.

Our research highlights the importance of integrating PDs into ZSQ methods and shows that relying solely on SD is not necessary. We do not claim that the PDs leading to performance gains are unrelated to the OD or do not overlap with OD. On the contrary, some PDs capture latent properties of OD that may be difficult to represent using current generation methods since PDs themself are real-world datasets with a similar complex distribution as OD. Moreover, PDs are publicly available and easy to obtain, and our proposed method provides a straightforward way to select the most suitable PD to enhance ZSQ while minimizing costs. In summary, our contributions are: 1) We introduce the incorporation of PDs into the current ZSQ pipeline to address the OD-lacking issue and provide a systematic understanding of the role of PDs in ZSQ, which may deliver insights to other data-free tasks. 2) We demonstrate that using PDs in ZeroP can be easily integrated into existing ZSQ methods without incurring additional costs and consistently improve their performance. Additionally, we propose a simple and effective method for guiding PD selection. 3) Our results demonstrate a new SOTA performance level in the ZSQ task and consistently outperform the SOTA methods on CIFAR10, CIFAR100, and ImageNet-1K. Specifically, on ImageNet-1K, ZeroP gains of 0.84% to 3.90% in 4-bit setting over 4 common architectures compared with SOTA methods. Besides, ZeroP achieves a comparable performance of OD methods, such as FDDA (Fang et al., 2019).

## 2 EXPLORING THE POTENTIAL OF PROXY DATA

The use of SD and PD to address the lack of OD has been studied for years, with early research dating back to Buciluǎ's work (Buciluǎ et al., 2006). At that time, OD often lay in a small submanifold of the attribute space. However, as modern datasets (Husain et al., 2019; Sun et al., 2017) grow larger and more complex, generating SD that accurately mimics the distribution of OD is becoming more challenging. While significant efforts have been devoted to developing SD methods that match the distribution of OD, PD has not received as much attention from the research community. One possible reason for this is that the complex distribution of modern datasets hinders the exploration of PD's potential.

Pure-SD methods (Cai et al., 2020; Xu et al., 2020; Zhong et al., 2022b) match OD by combining a set of discretely defined measures, such as BNS loss (Cai et al., 2020) and inter or intra-class loss (Zhong et al., 2022b). Although these methods may be suboptimal in approximating OD via limited measurements, they are more controllable than PD. Conversely, PD is

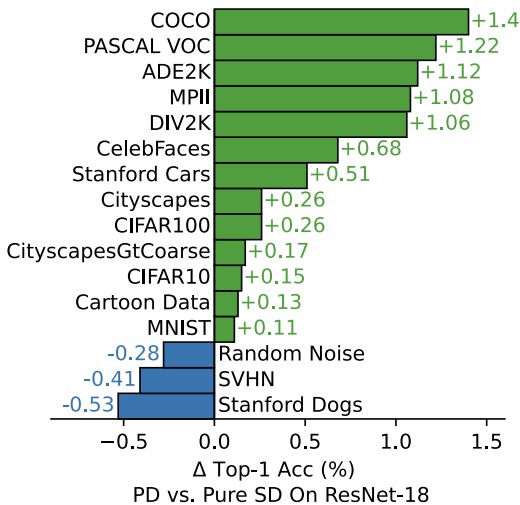

Figure 3: **Performance of 16 commonly used datasets as Proxy Data.** The top-1 accuracy deviation of the baseline and variants using 16 different PDs for image classification on ImageNet-1K as the OD in 4-bit setting.

a naturally occurring public dataset with a complex distribution, making PD's potential unclear. However, as the 4th column of Fig. 2 shows, PD, such as COCO (Lin et al., 2014), has a better t-SNE match for ImageNet-1K (Krizhevsky et al., 2017) than other SD generated by pure-SD methods, suggesting that PD may provide a promising approach to matching the rich latent features of large-scale datasets.

Recent works related PD, such as KnockoffNet (Orekondy et al., 2019) and DeGANs (Addepalli et al., 2020), are SD methods that use PD as guidance to generate SD and only test a limited number of PDs (#(PD) < 3) for data-free model stealing. It is, therefore, important to investigate whether PD has the potential to enhance performance of data-free tasks when used as a substitute for OD. To

this end, we constructed a simple baseline based on FDDA (Zhong et al., 2022a) using only SD and conducted extensive experiments by directly using a mixture of PD and SD as the input data for the quantization process. We considered 16 commonly used computer vision datasets with distinctive features as candidate PDs. Fig. 3 shows that while some PDs, such as SVHN (Netzer et al., 2011) and Stanford Dogs (Khosla et al., 2011), decreased accuracy compared to our manually constructed dataset Random Noise, 13 datasets increased performance, with more than 5 achieving a gain of 1.0% to 1.4% in accuracy. From this perspective, not only is PD readily accessible and often free, but also our results suggest that they offer a promising way to represent the latent rich features of OD with almost no cost. One remaining challenge is selecting appropriate and beneficial PDs from a large pool of PDs and we will present our method to address this challenge in the next section.

## 3 METHODOLOGY

### 3.1 PROBLEM FORMULATION

ZSQ aims to obtain a low-bit constraint network, *i.e.*, a quantized network $Q$, by leveraging information from a pre-trained full-precision model $F$, while working under the condition that the OD is either unavailable or hard to access. Typically, $F$ is trained on the task-specific dataset $D_{Ori} = \{(\boldsymbol{x}_i, y_i)\}$, and achieves the best performance on the original test dataset. For an input data $\boldsymbol{x}$, $F$ predicts a corresponding $K$-dimensional label vector $\boldsymbol{y} = F(\boldsymbol{x})$, where $\boldsymbol{y} \in [0, 1]^K$ and $\sum_k y_k = 1$. It is easy to convert $\boldsymbol{y}$ to a scalar class label $y$ and vice versa. In most ZSQ methods, the overall training loss $\mathcal{L}^{ZSQ}$ typically consists of two phases: updating the generator $G$ to generate SD and fine-tuning the quantized network $Q$. The corresponding loss functions are denoted by $\mathcal{L}^G$ and $\mathcal{L}^Q$, respectively. We will further explain $\mathcal{L}^G$ and $\mathcal{L}^Q$ in the following section.

**Data synthesis.** In ZSQ methods, when the OD is unavailable, the SD, $D_{Syn} = \{(\hat{\boldsymbol{x}}_i, \hat{y}_i)\}$, is typically generated using a generator $G$, where $\hat{\boldsymbol{x}}$ is produced by $G$ from a K-dimensional latent random vector $\boldsymbol{z}$ and a corresponding label $\hat{y}$. The producing process can be formulated as $\hat{\boldsymbol{x}} = G(\boldsymbol{z}|\hat{y}), \boldsymbol{z} \sim p(\boldsymbol{z})$, where $\boldsymbol{z}$ is sampled from a prior distribution $p(\boldsymbol{z})$, such as a Gaussian distribution $\mathcal{N}(0, 1)$. To ensure that the distribution of SD is as similar as possible to the OD, SD methods enforce the information of SD captured by the BNS in $F$ close to the information of OD captured by the same BNS. The loss $\mathcal{L}^G_{BNS}$ is defined as follows:

$$\mathcal{L}^G_{BNS} = \sum_{l=1}^{L} ||\boldsymbol{\mu}'_l(\hat{\boldsymbol{x}}) - \boldsymbol{\mu}^F_l||^2 + ||\boldsymbol{\sigma}'_l(\hat{\boldsymbol{x}}) - \boldsymbol{\sigma}^F_l||^2, \tag{1}$$

where $\boldsymbol{\mu}^F_l$, $\boldsymbol{\sigma}^F_l$ are the running mean and running variance at $l$-th layer of $F$. $\boldsymbol{\mu}'_l(\hat{\boldsymbol{x}})$, $\boldsymbol{\sigma}'_l(\hat{\boldsymbol{x}})$ are the mean and variance of $l$-th layer of $F$ for the input data $\hat{\boldsymbol{x}}$. $L$ is the total number of BN layers in $F$. In addition to $\mathcal{L}^G_{BNS}$, a cross-entropy loss term $\mathcal{L}^G_{CE}$ is also introduced to ensure that the generated data can be correctly identified by $F$. The loss $\mathcal{L}^G_{CE}$ is defined as follows:

$$\mathcal{L}^G_{CE} = \mathbb{E}_{(\boldsymbol{x},y) \sim \{(\hat{\boldsymbol{x}}, \hat{y})\}}[CE(F(\boldsymbol{x}), y)], \tag{2}$$

where $CE(\cdot, \cdot)$ is the cross-entropy loss function, and the expectation is taken over the pairs of inputs and labels over the SD. Finally, the overall loss function for updating $G$ is defined as $\mathcal{L}^G = \mathcal{L}^G_{BNS} + \alpha \mathcal{L}^G_{CE}$, where $\alpha$ is a hyperparameter that balances the importance of the two loss terms, $\mathcal{L}^G_{BNS}$ and $\mathcal{L}^G_{CE}$.

**Quantization with Synthetic Data.** When OD is unavailable, the quantized network $Q$ can be updated using the SD $D_{Syn}$ by optimizing the cross-entropy loss function as follows:

$$\mathcal{L}^Q_{CE} = \mathbb{E}_{(\boldsymbol{x},y) \sim \{(\hat{\boldsymbol{x}}, \hat{y})\}}[CE(Q(\boldsymbol{x}), y)]. \tag{3}$$

However, since the labels $\hat{y}$ in $D_{Syn}$ may be unreliable, ZSQ methods often use knowledge distillation (KD) (Hinton et al., 2015) to further force $Q$ to learn from the soft labels provided by the full-precision model $F$. The loss function for KD is defined as follows:

$$\mathcal{L}^Q_{KD} = \mathbb{E}_{(\boldsymbol{x},y) \sim \{(\hat{\boldsymbol{x}}, \hat{y})\}}[KL((Q(\boldsymbol{x}), F(\boldsymbol{x}))], \tag{4}$$

where, $KL(\cdot)$ is the Kullback-Leibler divergence. Therefore, the overall target optimization loss function for updating the quantized network $Q$ becomes $\mathcal{L}^Q = \mathcal{L}^Q_{CE} + \beta \mathcal{L}^Q_{KD}$, where $\beta$ is a hyper-parameter that balances the importance of the two loss terms, $\mathcal{L}^Q_{CE}$ and $\mathcal{L}^Q_{KD}$.

## 3.2 QUANTIZATION WITH PROXY DATA

**Proxy Data selection.** To choose an appropriate PD, we propose a simple and effective method based on the BNS. Specifically, we define a distance metric using BNS as follows:

$$\mathcal{D}_{BNS} = \frac{\sum_{m=1}^{M} \sum_{l=1}^{L} ||\boldsymbol{\mu}'_l(\tilde{\boldsymbol{x}}) - \boldsymbol{\mu}^F_l||^2 + ||\boldsymbol{\sigma}'_l(\tilde{\boldsymbol{x}}) - \boldsymbol{\sigma}^F_l||^2}{L}, \tag{5}$$

where, $\boldsymbol{\mu}^F_l$, $\boldsymbol{\sigma}^F_l$, and $L$ share the same definition as in Equ. 1. $\boldsymbol{\mu}'_l(\tilde{\boldsymbol{x}})$ and $\boldsymbol{\sigma}'_l(\tilde{\boldsymbol{x}})$ are the mean and variance calculated using the PD $\tilde{\boldsymbol{x}}$ instead of the SD $\hat{\boldsymbol{x}}$. Total $M$ samples are using to calculate the $\mathcal{D}_{BNS}$. A larger BNS distance indicates that the current PD is less related to the OD. To select an appropriate PD, we first calculate the BNS distance for all candidate PDs and rank them in decreasing order. We then select the PD with the smallest BNS distance, which is the one that is most closely related to the OD according to our distance metric. Note that choosing the PD with the smallest BNS distance is not always the best choice, as BNS distance is only a fast and cheap approximation of the true performance of a PD. However, it provides a reasonable starting point for selecting an appropriate PD.

**Using Proxy Data.** Instead of using PD as a guide to obtaining SD, we propose constructing the input data $\bar{\boldsymbol{x}}$ of ZeroP by combining PD $\tilde{\boldsymbol{x}}$ and SD $\hat{\boldsymbol{x}}$ as direct inputs for updating the quantized network. This approach allows the input data to benefit not only from the useful latent properties of PD but also from the specific features explicitly learned by SD. The input data $\bar{\boldsymbol{x}}$ can be obtained as follows:

$$\bar{\boldsymbol{x}} = Concat[\tilde{\boldsymbol{x}}_{1:\gamma B}, \hat{\boldsymbol{x}}_{\gamma B+1:B}], \tag{6}$$

where, $\gamma \in [0, 1]$ represents the mixing ratio of PD $\tilde{\boldsymbol{x}}$ and SD $\hat{\boldsymbol{x}}$. $\bar{\boldsymbol{x}}$ has a total batch size of $B$. $Concat[\cdot, \cdots, \cdot]$ module concatenates the PD $\tilde{\boldsymbol{x}}_{1:\gamma B}$ with $\gamma B$ batches and SD $\hat{\boldsymbol{x}}_{\gamma B+1:B}$ with $(1-\gamma)B$ batches, according to the batch dimension, to form the final input data $\bar{\boldsymbol{x}}$.

## 4 EXPERIMENTS

We conducted extensive experiments to demonstrate the efficiency and effectiveness of ZeroP. In this section, we provide an overview of the general experimental settings. Any specific changes will be mentioned in subsequent sections. We then present additional experiments to evaluate the benefits of PD and the relationship between BNS and performance, which support the research flow discussed in Sec. 2. Next, we compare the performance of ZeroP with SOTA methods. Additionally, we perform an ablation study on PD. Due to space limitations, we provide a detailed analysis and some experimental results in the supplemental materials (SM). We encourage readers to refer to the SM for a comprehensive understanding of PD and ZeroP.

**Datasets & Baselines.** To compare the performance of ZeroP with other SOTA methods, we report the top-1 accuracy on ImageNet-1K (Krizhevsky et al., 2017). Additionally, we evaluate the generalization of ZeroP by reporting the top-1 accuracy on CIFAR10 (Krizhevsky et al., 2009), CIFAR100 (Krizhevsky et al., 2009), and ImageNet-1K (Krizhevsky et al., 2017). Furthermore, we assess the performance gain of PD by using 16 datasets as PD, including CityscapesgtCoarse (Cordts et al., 2016), ADE2K (Zhou et al., 2017), Cartoon [1], DIV2K (Agustsson & Timofte, 2017), CIFAR10 (Krizhevsky et al., 2009), CIFAR100 (Krizhevsky et al., 2009), PASCAL VOC (Evering-ham et al., 2010), COCO (Lin et al., 2014), Cityscapes (Cordts et al., 2016), MNIST (LeCun et al., 1998), Random Noise [2], MPII (Andriluka et al., 2014), Stanford Dogs (Khosla et al., 2011), Stanford Cars (Krause et al., 2013), SVHN (Netzer et al., 2011), and CelebFaces (Liu et al., 2015). Please refer to the SM for the deatils of all the datasets.

---

[1] We will release the dataset after the paper accepted.

[2] Manually constructed by the authors.

Table 1: **The gain of 16 PDs on ImageNet-1K (OD).** The result of 4 commonly used architectures in 4-bit quantization is reported. 'FP32 Acc' denotes the full-precision model performances. PDs are divided into 3 groups based on the BNS distance for better display.

| BW | ProxyData | BN | ResNet-18 | MobileNetV1 | MobileNetV2 | RegNet-600MF |
|---|---|---|---|---|---|---|
| | FP32 Acc | 0.00 | 71.47 | 73.39 | 72.49 | 73.71 |
| | ZeroP$^{w/o}$ | - | 66.35 | 43.31 | 66.22 | 63.70 |
| | SVHN | 108.08 | 65.94 (0.41↓) | 46.66 (3.35↑) | 65.67 (0.55↓) | 64.13 (0.43↑) |
| | CIFAR10 | 82.14 | 66.50 (0.15↑) | 49.20 (5.89↑) | 66.60 (0.38↑) | 64.13 (0.43↑) |
| | CIFAR100 | 81.47 | 66.61 (0.26↑) | 50.35 (7.04↑) | 66.21 (0.01↓) | 64.73 (1.03↑) |
| | MNIST | 78.67 | 66.46 (0.11↑) | 47.36 (4.05↑) | 66.17 (0.05↓) | 63.17 (0.53↓) |
| | Random Noise | 78.09 | 66.06 (0.29↓) | 41.37 (1.94↓) | 65.65 (0.57↓) | 62.36 (1.34↓) |
| 4w4a | CityscapesgtCoarse | 41.76 | 66.52 (0.17↑) | 50.48 (7.17↑) | 66.16 (0.06↓) | 63.53 (0.17↓) |
| | Cartoon | 37.7 | 66.48 (0.13↑) | 50.03 (6.72↑) | 66.47 (0.25↑) | 64.65 (0.95↑) |
| | Cityscapes | 25.53 | 66.61 (0.26↑) | 49.54 (6.23↑) | 66.50 (0.28↑) | 64.50 (0.80↑) |
| | StanfordDogs | 15.02 | 65.82 (0.53↓) | 51.10 (7.79↑) | 66.79 (0.57↑) | 65.55 (1.85↑) |
| | StanfordCars | 13.45 | 66.86 (0.51↑) | 51.75 (8.44↑) | 66.67 (0.45↑) | 65.16 (1.46↑) |
| | CelebFaces | 11.92 | 67.03 (0.68↑) | 53.85 (10.54↑) | 67.11 (0.89↑) | 65.98 (2.28↑) |
| | ADE2K | 4.55 | 67.48 (1.13↑) | 56.40 (13.09↑) | 67.13 (0.91↑) | 66.79 (3.09↑) |
| | DIV2K | 4.20 | 67.41 (1.06↑) | 58.07 (14.76↑) | 67.76 (1.54↑) | 67.10 (3.40↑) |
| | MPII | 3.67 | 67.44 (1.09↑) | 56.61 (13.30↑) | 67.24 (1.02↑) | 66.65 (2.95↑) |
| | PASCALVOC | 2.18 | 67.52 (1.17↑) | 59.12 (15.81↑) | 67.87 (1.65↑) | 67.32 (3.62↑) |
| | COCO | **2.17** | **67.75** (1.40↑) | **59.38** (16.07↑) | **68.13** (1.91↑) | **67.92** (4.22↑) |

We conduct experiments on several commonly used network architectures, including ResNet-18 (He et al., 2016), ResNet-20 (He et al., 2016), ResNet-50 (He et al., 2016), MobileNetV1 (Howard et al., 2017), MobileNetV2 (Sandler et al., 2018), and RegNet-600MF (Radosavovic et al., 2020). In specific experiments, we may focus on a subset of these architectures, and we will provide detailed settings in the corresponding sections. As for the baselines, we compare our results with current SOTA ZSQ methods using SD, such as GDFQ (Xu et al., 2020), Qimera (Choi et al., 2021), and IntraQ (Zhong et al., 2022b). All pre-trained full-precision models are sourced from the PyTorchCV library, and all experiments are implemented using PyTorch (Paszke et al., 2019) and run on A100 GPUs.

**Implementation details.** We use the original open-source configurations for the baselines we used or mentioned in our experiment, such as GDFQ, Qimera, IntraQ, and FDDA (Zhong et al., 2022a). Moreover, we use a '*' to highlight our implementation method or results based on the official code. We set $\gamma = 0.5$ and COCO as PD for ImageNet-1K and DIV2K as PD for CIFAR10/CIFAR100 for default. Furthermore, we implement our ZeroP based on the FDDA framework. We keep the same configuration of FDDA in open source code[3]. For the loss, we only use loss mentioned in Sec. 3, i.e., $\mathcal{L}^G$ for updating the generator $G$, and $\mathcal{L}^Q$ for fine-tuning the quantized network $Q$. Any changes to specific configurations will be mentioned in the corresponding experiment section. Due to space limitations, please refer to the SM for more details.

### 4.1 EXPLORING THE GAINS OF PROXY DATA

### 4.1.1 ADDITIONAL RESULTS ON PROXY DATA

**Settings.** In Sec. 2, we emphasized the importance of exploring the potential of PD for ZSQ. Here, we present additional experiment results that provide more details on the gains of using PD. To obtain a comprehensive analysis, we construct a simple ZeroP using only SD as a baseline and 16 different datasets as PD to the baseline. For brevity, we use ZeroP$^{w/o}$ and ZeroP$^{w/}$ to indicate ZeroP using only SD and ZeroP using PD, respectively. We evaluate ZeroP$^{w/}$ on ResNet-18, MobileNetV1, MobileNetV2, and RegNet-600MF in 4-bit (4w/4a) and 5-bit (5w/5a) quantization settings, and report the top-1 accuracy on the ImageNet-1K validation set in Tab. 1. We manually construct a 'Random Noise' dataset via sampling the image value from a Gaussian distribution $\mathcal{N}(0, 1)$ with the same size of SD/PD/OD. Due to space limitations, we only list the 4-bit experiment results in the table. Please refer to the SM for the 5-bit experiment results.

---

[3]https://github.com/zysxmu/FDDA

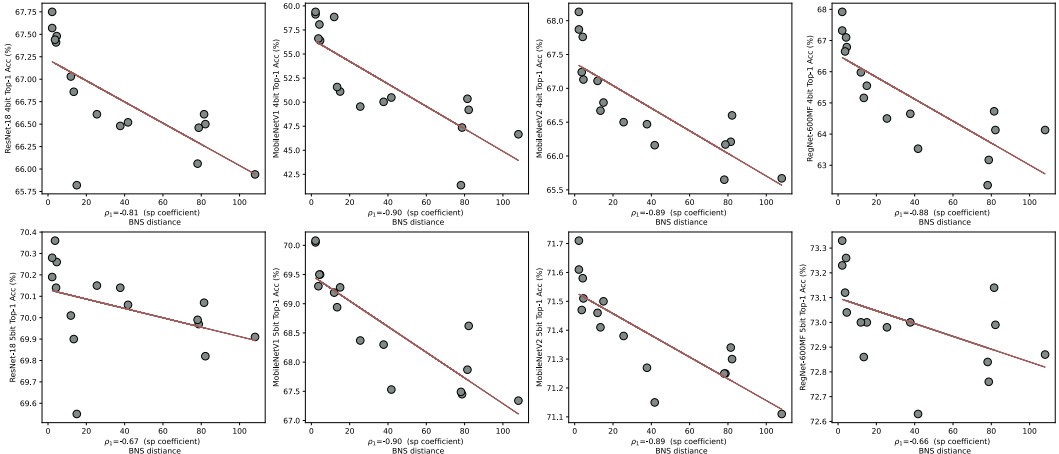

Figure 4: **The correlation of BNS distance and accuracy.** Spearman's rank correlation coefficient, $\rho_1$, measures the fidelity of BNS distance for choosing an appropriate PD. The figure shows the $\rho_1$ for 4 architectures on 4-bit and 5-bit setting with ZeroP as baseline.

**Results.** Tab. 1 shows that most cases where the performance decreases occur when the BNS distance is large, such as in the cases of SVHN, CityscapesgtCoarse, and MNIST. However, using PDs with small BNS distance, *e.g.*, COCO and ADE2K, consistently improved the baseline significantly, with approximately 1.40%, 10.00%, 1.50%, and 3.00% performance gain in accuracy for ResNet-18, MobileNetV1, MobileNetV2, and RegNet-600MF, respectively. Interestingly, some datasets, *e.g.* CIFAR10, have larger BNS distances than Random Noise, yet they still yield better performance. Also, one may notice that some PDs, such as SVHN and MNIST, exhibit worse performance than Random Noise. We speculate that this decrease in performance for SVHN and MINIST may be due to their domains being farther away from the ImageNet-1K domain. In summary, our experimental results demonstrate that using PDs can lead to significant performance gains, even in different low-bit cases on large-scale datasets like ImageNet-1K.

### 4.1.2 SELECTING PROXY DATA WITH BNS

Now that we have shown that PDs can improve performance, it is important to find a way to reduce the cost of selecting the appropriate PD for specific tasks. Running ZSQ with all possible candidate PDs can be prohibitively expensive in practice, given the large number of PDs available. Here, we demonstrate the effectiveness of the proposed selection method with experiments.

**Settings.** We randomly selected 1024 images, $M = 1024$ in Equ. 5, from a specific PD and calculated the BNS distance using Equ. 5. We calculated the BNS distance for 16 PDs, and the results are listed in Tab. 1. To better visualize the results, we divided the 16 datasets into three groups based on their BNS distance: less than 10, 10 to 50, and greater than 50. We also calculated the Spearman's rank correlation coefficient ($\rho_1$) and Pearson correlation coefficient ($\rho_2$) to measure the relationship between the BNS distance and the final performance. All experiments were performed on the ImageNet-1K dataset. Due to space limitations, the results of Pearson correlation coefficient are list in SM. Please refer to SM for more details.

**Results.** As shown in Tab. 1, we observed that smaller BNS distance is associated with better performance gain for PDs. We also found that performance degradation of PDs only occurred in large BNS distance datasets such as Stanford Dogs and SVHN. Furthermore, as shown in Fig. 4, the BNS distance is highly related to the final performance, which is reasonable since BNS is widely used in SD methods as the main guideline to extract the features of OD into SD. Additionally, Zhang's work (Zhang et al., 2021a) provides more experiments to explain how BN contains information on OD.

## 4.2 PERFORMANCE COMPARISON

**Settings.** This study aims to compare the performance of ZeroP with SOTA methods in 4-bit and 5-bit quantization settings for the ZSQ task. We report the top-1 accuracy on the validation set of ImageNet-1K and compare it with the current SOTA methods. These SOTA methods can be further divided into two groups: 1) ZSQ methods that use OD to accomplish quantization, including ACIQ-Mix (Banner et al., 2019), AdaQuant (Hubara et al., 2020), Bit-Split (Wang et al., 2020), BERCQ (Li et al., 2021), and FDDA (Zhong et al., 2022a). OD methods usually have higher performance compared to SD methods. 2) ZSQ methods that use only SD, such as HAST (Li et al., 2023), AIT (Choi et al., 2022), IntraQ (Zhong et al., 2022b), Qimera (Choi et al., 2021), and GDFQ (Xu et al., 2020). We list the OD methods to show the performance upper bound of the ZSQ task. All results are shown in Tab. 2.

**Results.** Tab. 2 shows that ZeroP, with SD and PD, outperforms almost all pure SD methods except for AIT (71.96%) in the 5-bit case. In the 5-bit case, all the methods can better approach the full-precision model (FP32 Acc) for the relaxed bit width constraint, so the difference between methods may not be distinctive. However, the superiority of ZeroP is fully demonstrated in the 4-bit case. One may observe that ZeroP achieves performance increases of 0.84%, 3.90%, 1.68%, and 1.32% for ResNet-18, ResNet-50, MobileNetV1, and MobileNetV2, respectively, in comparison with the highest-performing solo SD methods. ZeroP consistently outperforms all solely SD SOTA methods. Meanwhile, FDDA remains almost the highest accuracy over all methods since it utilizes OD. ZeroP still outperforms many OD methods in 4-bit and 5-bit cases, as shown in Tab. 2. We also conduct a performance comparison on CIFAR10/CIFAR100. Please refer to the SM for more details.

Table 2: **Image classification performance comparisons with SOTA methods on ImageNet-1K.** 'BW' means Bit-Width. This table report 4-bit and 5-bit quantization settings. '*' denotes our implementation results based on the official code.

| BW | Methods | Dataset Type | ResNet-18 | ResNet-50 | MobileNetV1 | MobileNetV2 |
|---|---|---|---|---|---|---|
| | FP32 Acc | – | 71.47 | 77.73 | 73.39 | 72.49 |
| 5w5a | ACIQ-Mix | OD | 66.80 | – | 51.65 | 60.42 |
| | AdaQuant | OD | 68.19 | – | – | 63.61 |
| | Bit-Split | OD | 68.88 | – | – | – |
| | BRECQ | OD | 70.27 | – | 66.51 | 70.26 |
| | FDDA | OD | 70.56 | – | – | 71.63 |
| | GDFQ | SD | 68.24* | 71.90* | 59.34* | 67.85* |
| | Qimera | SD | 69.29 | 75.32 | 61.89* | 70.45 |
| | IntraQ | SD | 69.94 | 74.64* | 68.17 | 71.28 |
| | AIT | SD | 70.28 | 76.00 | - | **71.96** |
| | HAST | SD | – | – | 68.52 | 71.72 |
| | ZeroP$^{w/o}$ | SD | 69.92 | 75.36 | 67.30 | 71.23 |
| | ZeroP$^{w/}$ | SD+PD | 70.28 | 76.10 | 70.08 | 71.71 |
| 4w4a | ACIQ-Mix | OD | 57.47 | – | 4.68 | 34.84 |
| | AdaQuant | OD | 63.45 | – | – | 34.64 |
| | Bit-Split | OD | 67.49 | – | – | – |
| | BRECQ | OD | 67.94 | – | 57.11 | 63.64 |
| | FDDA | OD | 68.88 | – | – | 63.75 |
| | GDFQ | SD | 60.60 | 56.04* | 30.79* | 59.56* |
| | Qimera | SD | 63.84 | 66.25 | 38.66* | 61.62 |
| | IntraQ | SD | 66.47 | 56.88* | 51.36 | 65.10 |
| | AIT | SD | 66.83 | 68.27 | - | 66.81 |
| | HAST | SD | 66.91 | – | 57.70 | 65.60 |
| | ZeroP$^{w/o}$ | SD | 66.35 | 64.50 | 43.31 | 63.70 |
| | ZeroP$^{w/}$ | SD+PD | **67.75** | **72.17** | **59.38** | **68.13** |

Table 3: **The generalization of ZeroP and the ablation study on PD.** Where 'RN' indicates 'Random Noise'. 'SD', 'PD', 'OD', and 'BW' share the same defination as before Tables. For each method, every 'SD' and 'PD' columns can be consider as showing the generalization of ZeroP. For each method, 'RN', 'SD', 'PD', and 'OD' can be viewed as the ablation on PD.

| Dataset | Model (FP32 Acc) | BW | GDFQ +SD | GDFQ +RN | GDFQ +OD | GDFQ +PD | Qimera +SD | Qimera +RN | Qimera +OD | Qimera +PD | IntraQ +SD | IntraQ +RN | IntraQ +OD | IntraQ +PD | ZeroP +SD | ZeroP +RN | ZeroP +OD | ZeroP +PD |
|---|---|---|---|---|---|---|---|---|---|---|---|---|---|---|---|---|---|---|
| CIFAR10 | ResNet-20 94.03 | 5w5a | 93.39* | 92.13 | 93.85 | 93.60(0.21↑) | 93.46 | 92.50 | 93.68 | **93.77**(0.31↑) | 93.28* | 93.33 | 93.40 | 93.35(0.07↑) | 93.72 | 93.49 | 93.91 | 93.73(0.01↑) |
| | | 4w4a | 90.25 | 88.04 | 93.11 | 92.53(2.28↑) | 91.26 | 88.70 | 92.94 | 92.81(1.55↑) | 91.49 | 90.76 | 91.52 | 91.84(0.35↑) | 92.24 | 91.40 | 93.17 | **93.00**(0.76↑) |
| | | 3w3a | 70.98* | 67.34 | 89.01 | 87.15(16.17↑) | 77.64* | 73.43 | 89.43 | 87.48(9.84↑) | 77.07 | 75.88 | 87.94 | 85.63(8.56↑) | 79.43 | 78.23 | 89.25 | **88.24**(8.81↑) |
| CIFAR100 | ResNet-20 70.33 | 5w5a | 67.45* | 65.55 | 68.71 | 68.04(0.59↑) | 69.02 | 66.41 | 69.86 | **69.56**(0.54↑) | 68.17* | 67.92 | 69.51 | 68.66(0.49↑) | 69.53 | 68.75 | 70.02 | 69.52(0.01↓) |
| | | 4w4a | 63.80 | 57.61 | 66.98 | 66.31(2.51↑) | 65.10 | 61.62 | 68.10 | 67.41(2.31↑) | 64.98 | 64.74 | 66.80 | 66.15(1.17↑) | 66.76 | 65.51 | 68.55 | **67.64**(0.88↑) |
| | | 3w3a | 49.62* | 36.19 | 58.46 | 55.40(5.78↑) | 47.44* | 43.84 | 60.76 | 56.32(8.88↑) | 48.25 | 48.45 | 56.26 | 53.68(5.43↑) | 52.26 | 52.93 | 60.65 | **57.46**(5.20↑) |
| ImageNet-1K | ResNet-18 71.47 | 5w5a | 68.24* | 68.37 | 69.29 | 69.27(1.03↑) | 69.29 | 68.98 | 70.10 | 70.12(0.83↑) | 69.94 | 69.98 | 70.31 | 70.24(0.30↑) | 69.92 | 69.99 | 70.44 | **70.28**(0.36↑) |
| | | 4w4a | 60.60 | 60.51 | 66.05 | 62.50(1.90↑) | 63.84 | 63.19 | 66.99 | 66.52(2.68↑) | 66.47 | 65.56 | 67.89 | 67.67(1.20↑) | 66.35 | 66.06 | 67.81 | **67.75**(1.40↑) |
| | ResNet-50 77.73 | 5w5a | 71.90* | 72.86 | 75.53 | 75.59(3.69↑) | 75.32 | 73.77 | 75.53 | 76.06(0.74↑) | 74.64* | 73.78 | 75.99 | 75.83(1.19↑) | 75.34 | 74.95 | 76.41 | **76.10**(0.74↑) |
| | | 4w4a | 56.04* | 50.85 | 70.92 | 65.98(9.94↑) | 66.25 | 61.71 | 69.71 | 68.62(2.37↑) | 56.88* | 59.16 | 69.78 | 70.88(14.00↑) | 64.50 | 66.60 | 72.44 | **72.17**(7.67↑) |
| | MobileNetV1 73.39 | 5w5a | 59.34* | 59.54 | 66.92 | 65.02(5.68↑) | 61.89* | 63.97 | 69.74 | 69.11(7.22↑) | 68.17 | 67.70 | 69.87 | 69.81(1.64↑) | 67.30 | 67.49 | 70.06 | **70.08**(2.78↑) |
| | | 4w4a | 30.79* | 30.05 | 53.24 | 38.41(7.62↑) | 38.66* | 41.58 | 57.80 | 53.00(14.34↑) | 51.36 | 49.54 | 59.66 | 57.21(5.85↑) | 43.31 | 41.37 | 61.07 | **59.38**(16.07↑) |
| | MobileNetV2 72.49 | 5w5a | 67.85* | 68.22 | 70.96 | 70.14(2.29↑) | 70.45 | 69.46 | 71.70 | 71.48(1.03↑) | 71.28 | 71.24 | 72.10 | 71.79(0.51↑) | 71.23 | 71.25 | 71.70 | **71.71**(0.48↑) |
| | | 4w4a | 59.56* | 59.55 | 66.03 | 64.63(5.07↑) | 61.72 | 59.39 | 66.74 | 65.72(4.00↑) | 65.10 | 65.72 | 67.90 | 67.73(2.63↑) | 63.70 | 65.65 | 68.38 | **68.13**(4.43↑) |

## 4.3 ABLATION STUDY AND ANALYSIS

**Settings.** In this section, we perform an ablation study to evaluate the effectiveness of PD used in ZeroP on CIFAR10, CIFAR100, and ImageNet-1K. Specifically, we construct control experiments as follows. We first construct each baseline method's input data as defined in Equ. 6. Then, we compare four different input data: pure-SD ('SD'), mixed with Random Noise ('RN'), mixed with Original Data ('OD'), and mixed with Proxy Data ('PD'). We set $\gamma = 0.5$ for all mixed input data cases, and the Random Noise for 'RN' is constructed as follows: we directly sample the value of

RN image from a Gaussian distribution $\mathcal{N}(0,1)$ with as the same size as SD/PD/OD. We test the following baseline methods: GDFQ, Qimera, IntraQ, and ZeroP. The results are reported in Tab. 3.

**Results.** As shown in Tab. 3, for each baseline method, we roughly have 'OD' > 'PD' > 'SD' > 'RN'. Still, there are some counterexamples, such as 'PD' > 'OD' (ResNet-20 of Qimera on CIFAR10 in 5-bit case) and 'RN' > 'SD' (MobileNetV1 of Qimera on ImageNet-1K). This result is consistent with our expectations. 'OD' represents the performance upper bound, and 'PD' and 'SD' is approximate 'OD'. With appropriate PD, 'PD' is better than 'SD' since 'SD' may only provide limited single-angle mimicry of 'OD'. Finally, 'RN' may contain useless or even harmful information, which results in almost the worst performance. This phenomenon is observed almost consistently across all baseline methods and datasets. The result of Tab. 3 also demonstrates the ZeroP pipeline can easily transfer to other SOTA pure-SD ZSQ methods. Besides, due to the limited space, the extensive analysis of PD is presented in SM. Please refer to SM for more details.

## 5 RELATED WORK

**Data-free setting tasks.** Data-free settings have been widely studied in various tasks, such as data-free Knowledge Distillation (KD) (Lopes et al., 2017; Chen et al., 2019; Nayak et al., 2019; Fang et al., 2019; Liu et al., 2021) and Model Stealing (MS) (Tramèr et al., 2016; Wang & Gong, 2018; Oh et al., 2019; Sanyal et al., 2022; Barbalau et al., 2020; Orekondy et al., 2019). Data-free KD typically assumes a "Teacher" and "Student" model relationship, aiming to transfer teacher knowledge to the student model. For example, Lopes et al. (2017) propose utilizing metadata to achieve data-free KD. However, the reliance on metadata does not fully match the concept of a total data-free setting. Other works (Chen et al., 2019; Oh et al., 2019; Fang et al., 2019) further propose obtaining synthetic data during distillation, eliminating the need for OD. On the other hand, Model Stealing aims to extract specific information from the victim network, such as hyperparameters (Wang & Gong, 2018), functionality (Tramèr et al., 2016; Barbalau et al., 2020; Orekondy et al., 2019), or architecture (Oh et al., 2019), often assuming limited access to the victim through certain APIs. Recent works explore stronger constraints, such as complete black-box victim models (Barbalau et al., 2020) or hard-label settings (Sanyal et al., 2022). Data-free KD and ZSQ are, to some extent, more relaxed compared to complete black-box data-free tasks like MS (Barbalau et al., 2020; Orekondy et al., 2019) since the teacher (victim) model's information is available in ZSQ and data-free KD. From this perspective, exploring Proxy Data in our ZeroP approach may benefit other data-free setting tasks.

## 6 LIMITATIONS

Although ZeroP outperforms current pure-SD methods by a large margin and achieves comparable performance to OD methods, some potential insufficiencies should be considered: 1) Obtaining PD can be challenging for certain tasks. In some cases, there may be no available candidate datasets to serve as PD. 2) We only experimented with the BNS distance selection method on CNN networks. More recent architectures, such as Vision Transformer (Dosovitskiy et al., 2020), have yet to be tested due to resource limitations. 3) Alternative PD selection methods should be explored, since Batch normalization (BN) may not suit some task. For example, in image super-resolution (Ledig et al., 2017), BN may be harmful to performance.

## 7 CONCLUSION

In this paper, we propose ZeroP, a novel approach for the ZSQ task. We investigate the potential gain of PD across 16 commonly used CV datasets. Additionally, we introduce the BNS distance as a simple yet effective metric for selecting suitable PD for a specific task. Unlike previous works that rely solely on SD or use PD to guide SD generation, ZeroP directly combines PD and SD as input for the ZSQ task. Experiment results demonstrate that ZeroP outperforms SOTA pure-SD methods by a large margin on diverse datasets. Furthermore, ZeroP achieves competitive performance compared to existing OD methods. The performance gain of ZeroP is consistent across various SOTA pure-SD methods. Our findings challenge the notion that relying solely on SD is necessary.

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
