

Figure 5: **The examples of 16 computer vision datasets.** We will release the 'Cartoon' dataset soon. And the 'Random Noise' dataset is constructed via sampling the image value from a Gaussian distribution $\mathcal{N}(0, 1)$ with the same size of SD/PD/OD. 'CityGTCoarse' means the CityScapesGT-Coarse dataset for brevity.

## A  EXPERIMENTAL DETAILS

### A.1  DATASETS

**ImageNet-1K.**  The ImageNet-1K (Krizhevsky et al., 2017) dataset is a large-scale database with 1,000 object classes and over one million images. Each image has high-quality annotation information for classification and localization and covers many different types of images. This dataset has been widely used in computer vision research fields such as image classification, object detection, and scene understanding. It is an essential foundation for computer vision research.

**COCO.**  The COCO (Lin et al., 2014) dataset contains over 330K images and over 2.5M instances for object detection, instance segmentation, and keypoint detection. The images come from diverse sources and are accurately annotated, with each instance having a pixel-level ground truth. COCO is one of the most representative computer vision datasets, making significant contributions to easy-to-use data availability and facilitating deep learning and advanced visual algorithm research.

**Stanford Dogs.**  The Stanford Dogs (Khosla et al., 2011) dataset contains images of 120 breeds of dogs from around the world. This dataset has been built using images and annotation from ImageNet for the task of fine-grained image categorization. The dataset contains 120 categories, with a total of 20,580 images collected.

**Stanford Cars.**  The Stanford Cars (Krause et al., 2013) dataset provides a collection of car images suitable for use in computer vision research, consisting of 16,185 images of 196 different car classes. Each image is labeled with a corresponding image-level label and bounding box coordinates annotation. The dataset has been widely used for research related to automobile image classification, detection, and fine-grained recognition, and corresponding competitions have been established for model evaluation.

**Cartoon.**  The Cartoon dataset contains 21,551 image with cartoon head sculpture. We will release it soon.

**SVHN.** The SVHN (Netzer et al., 2011) is a real-world image dataset for developing machine learning and object recognition algorithms with minimal requirements on data preprocessing and formatting. It can be seen as similar in flavor to MNIST (e.g., the images are of small cropped digits) but incorporates an order of magnitude more labeled data (over 600,000 digit images) and comes from a significantly harder, unsolved, real-world problem (recognizing digits and numbers in natural scene images). SVHN is obtained from house numbers in Google Street View images.

**CelebFaces.** The CelebFaces (Liu et al., 2015) Attributes Dataset (CelebA) is a large-scale face attributes dataset with more than 200K celebrity images, each with 40 attribute annotations. The images in this dataset cover large pose variations and background clutter. CelebA has large diversities, large quantities, and rich annotations, including 10,177 identities,202,599 face images, 5 landmark locations, and 40 binary attribute annotations per image.

**MPII.** The MPII (Andriluka et al., 2014) Human Pose dataset includes 24,953 training images and 6,990 testing images for human pose estimation. Each person in each image is annotated with 14 body key points. The dataset has been widely used to study algorithms for human pose estimation, and its performance has become one of the standards for model evaluation in this field.

**ADE20K.** The ADE20K (Zhou et al., 2017) dataset is a large-scale dataset for scene semantic segmentation. The dataset comprises 25,000 annotated images and 150 scene categories for training, testing, and validation purposes. Specifically, the training set contains 20,210 images, the validation set includes 2,000 images, and the test set has 2,000 images.

**CIFAR10.** The CIFAR10 (Krizhevsky et al., 2009) dataset consists of 60,000 $32 \times 32$ color images in 10 classes, with 6,000 images per class. There are 50,000 training images and 10,000 test images. The dataset is divided into five training batches and one test batch, each with 10,000 images. The test batch contains exactly 1,000 randomly-selected images from each class. The training batches contain the remaining images in random order, but some training batches may contain more images from one class than another. Between them, the training batches contain exactly 5,000 images from each class.

**CIFAR100.** The CIFAR100 (Krizhevsky et al., 2009) is just like the CIFAR10, except it has 100 classes containing 600 images each. There are 500 training images and 100 testing images per class. The 100 classes in the CIFAR-100 are grouped into 20 superclasses. Each image comes with a "fine" label (the class to which it belongs) and a "coarse" label (the superclass to which it belongs).

**Cityscapes.** The Cityscapes (Cordts et al., 2016) is a new large-scale dataset that contains a diverse set of stereo video sequences recorded in street scenes from 50 different cities, with high-quality pixel-level annotations of 5,000 frames in addition to a larger set of 20,000 weakly annotated frames. The dataset is thus an order of magnitude larger than similar previous attempts.

**CityScapesGTCoarse.** The CityScapesGTCoarse (Cordts et al., 2016) dataset is a subset of the Cityscapes dataset, which mainly includes coarse semantic labels of 19 semantic categories at different resolutions. This dataset was created to improve the performance of traditional semantic segmentation models in pixel-level details compared to the Fine data in the Cityscapes dataset. The labels in the GTCoarse data are coarser than those in the Cityscapes Fine data. The GTCoarse dataset contains 5,000 training images, 1,000 validation images, and 1,500 testing images, each with corresponding semantics and bounding box json files. The CityScapes GTCoarse dataset is widely used in research work related to urban scene segmentation.

**DIV2K.** The DIV2K (Agustsson & Timofte, 2017) consists of RGB images with a wide range of contents. The dataset is divided into three subsets: training data, validation data, and test data. The training data includes 800 high-resolution images with corresponding low-resolution images provided for 2, 3, and 4 downscaling factors. For validation data, 100 high-resolution images are used for generating corresponding low-resolution images.

**MNIST.** The MNIST (LeCun et al., 1998) database of handwritten digits has a training set of 60,000 examples and a test set of 10,000 examples. It is a subset of a larger set available from NIST. The digits have been size-normalized and centered in a fixed-size image.

**PASCAL VOC.** The PASCAL VOC (Everingham et al., 2010) dataset aims for visual object detection and segmentation. The dataset comprises 20 object categories and 10,064 images, each with a complete pixel-level annotation. The primary use of PASCAL VOC is to study algorithms for visual object recognition and segmentation. Additionally, evaluation methods and benchmark results are provided. PASCAL VOC is widely used in computer vision and is considered as an important reference for numerous research and technological advancements.

**Random Noise.** The Random Noise[4] dataset is constructed via sampling the image value from a Gaussian distribution $\mathcal{N}(0, 1)$ with the size of SD/PD/OD.

## A.2 IMPLEMENT DETAILS

**IntraQ+PD.** To generate the data, we directly import the generator from IntraQ (Zhong et al., 2022b), the Adam (Kingma & Ba, 2014) optimizer with a momentum of 0.9 and an initial learning rate of 0.5. We update the synthesized images for 1,000 iterations and decay the learning rate by 0.1 when the loss has not decreased for 50 consecutive iterations. The batch size is set to 256 for all datasets. We generated 5,120 images according to the experimental configuration of IntraQ. We fine-tune the quantized model and calculate the SGD using the Nesterov method with a momentum of 0.9 and weight decay of $10^{-4}$. We set the batch size to 25 for CIFAR10/CIFAR100 and 16 for the ImageNet-1K dataset. We set the initial learning rates to $10^{-5}$ and $10^{-6}$ for CIFAR10 and CIFAR100 and ImageNet-1K, respectively, and decay these learning rates by 0.1 every 100 fine-tuning epochs for a total of 150 epochs. We construct input data as defined in Equ. 6,We set $\gamma = 0.5$, and COCO as PD for ImageNet-1K and DIV2K as PD for CIFAR10 and CIFAR100 for default.

**GDFQ+PD.** We directly import the generator from GDFQ (Xu et al., 2020) to generate the data,For CIFAR10 and CIFAR100,we constructed a generator based on ACGAN (Odena et al., 2017) and added random Gaussian noise. During training, we optimized the generator and quantized model using both Adam and SGD with Nesterov (Nesterov, 1983), where the momentum coefficient and weight decay were set to 0.9 and $10^{-4}$, respectively. In addition, we initialized the learning rates of the quantized model and generator to $10^{-4}$ and $10^{-3}$, respectively. They were both decayed by 0.1 every 100 epochs. We trained the generator and quantized model for 400 epochs with 200 iterations per epoch. To obtain a more stable range of quantized activation values, we calculated the moving average of the activation range for the first four epochs without updating the quantized model and then fixed this range for subsequent training. For ImageNet-1K, we replace the standard batch normalization layer of the generator with a conditional batch normalization layer for classification after fusion with SN-GAN (Miyato et al., 2018), and set the initial learning rate of the quantized model to $10^{-6}$. Other training settings are the same as those for CIFAR10 and CIFAR100. We construct input data as defined in Equ. 6,We set $\gamma = 0.5$, and COCO as PD for ImageNet-1K and DIV2K as PD for CIFAR10 and CIFAR100 for default.

**ZeroP and Qimera+PD.** For the ZeroP method, We align with FDDA (Zhong et al., 2022a) for the basic experimental configuration. Specifically, we set the initial learning rates of the generator and the quantization network to $10^{-3}$ and $10^{-6}$, respectively. For the generator, we use Adam optimizer with a momentum of 0.9, and the learning rate is multiplied by 0.1 every 100 epochs. For the quantization network, we use SGD optimizer with Nesterov and weight decay of $10^{-4}$, and adjust the learning rate using cosine annealing (Loshchilov & Hutter, 2016). Before formal training, we set a 50-epoch warming-up update for G. Then, we use a total of 350 epochs to update the generator G and quantization model Q. Compared with FDDA, we do not calculate BNS distortion loss based on OD and do not need to calculate the BN center for each category of the real dataset. For the Qimera+PD method, We align with the ZSQ method in Qimera (Choi et al., 2021) for quantized pipeline and align with the basic experimental configuration of GDFQ. We construct input data in

---

[4]Manually constructed by the authors.

the above two methods as defined in Equ. 6.We set $\gamma = 0.5$, and COCO as PD for ImageNet-1K and DIV2K as PD for CIFAR10 and CIFAR100 for default.

Table 4: **The result of 4 commonly used architectures in 5-bit quantization is reported.** 'FP32 Acc' denotes the full-precision model performances. PDs are divided into 3 groups based on the BNS distance for better display.

| BW | ProxyData | BN | ResNet-18 | MobileNetV1 | MobileNetV2 | RegNet-600MF |
|---|---|---|---|---|---|---|
| | FP32 Acc | 0.00 | 71.47 | 73.39 | 72.49 | 73.71 |
| | ZeroP$^{w/o}$ | - | 69.92 | 67.30 | 71.23 | 72.76 |
| 5w5a | SVHN | 108.08 | 69.91 (0.01↓) | 67.34 (0.04↑) | 71.11 (0.12↓) | 72.87 (0.11↑) |
| | CIFAR10 | 82.14 | 69.82 (0.10↓) | 68.62 (1.32↑) | 71.30 (0.07↑) | 72.99 (0.23↑) |
| | CIFAR100 | 81.47 | 70.07 (0.15↑) | 67.87 (0.57↑) | 71.34 (0.11↑) | 73.14 (0.38↑) |
| | MNIST | 78.67 | 69.97 (0.05↑) | 67.45 (0.15↑) | 71.25 (0.02↑) | 72.76 (0.00) |
| | RandomNoise | 78.09 | 69.99 (0.07↑) | 67.49 (0.19↑) | 71.25 (0.02↑) | 72.84 (0.08↑) |
| | CityscapesgtCoarse | 41.76 | 70.06 (0.14↑) | 67.53 (0.23↑) | 71.15 (0.08↓) | 72.63 (0.13↓) |
| | Cartoon | 37.72 | 70.14 (0.22↑) | 68.30 (1.0↑) | 71.27 (0.04↑) | 73.00 (0.24↑) |
| | Cityscapes | 25.53 | 70.15 (0.23↑) | 68.37 (1.07↑) | 71.38 (0.15↑) | 72.98 (0.22↑) |
| | StanfordDogs | 15.02 | 69.55 (0.37↓) | 69.28 (1.98↑) | 71.50 (0.27↑) | 73.00 (0.24↑) |
| | StanfordCars | 13.45 | 69.90 (0.02↓) | 68.94 (1.64↑) | 71.41 (0.18↑) | 72.86 (0.10↑) |
| | CelebFaces | 11.92 | 70.01 (0.09↑) | 69.19 (1.89↑) | 71.46 (0.23↑) | 73.00 (0.24↑) |
| | ADE2K | 4.55 | 70.26 (0.34↑) | 69.50 (2.20↑) | 71.51 (0.28↑) | 73.04 (0.28↑) |
| | DIV2K | 4.20 | 70.14 (0.22↑) | 69.50 (2.20↑) | 71.58 (0.35↑) | 73.26 (0.50↑) |
| | MPII | 3.67 | **70.36** (0.44↑) | 69.30 (2.00↑) | 71.47 (0.24↑) | 73.12 (0.36↑) |
| | PASCALVOC | 2.18 | 70.19 (0.27↑) | 70.05 (2.75↑) | 71.61 (0.38↑) | 73.23 (0.47↑) |
| | COCO | **2.17** | 70.28 (0.36↑) | **70.08** (2.78↑) | **71.71** (0.48↑) | **73.33** (0.57↑) |

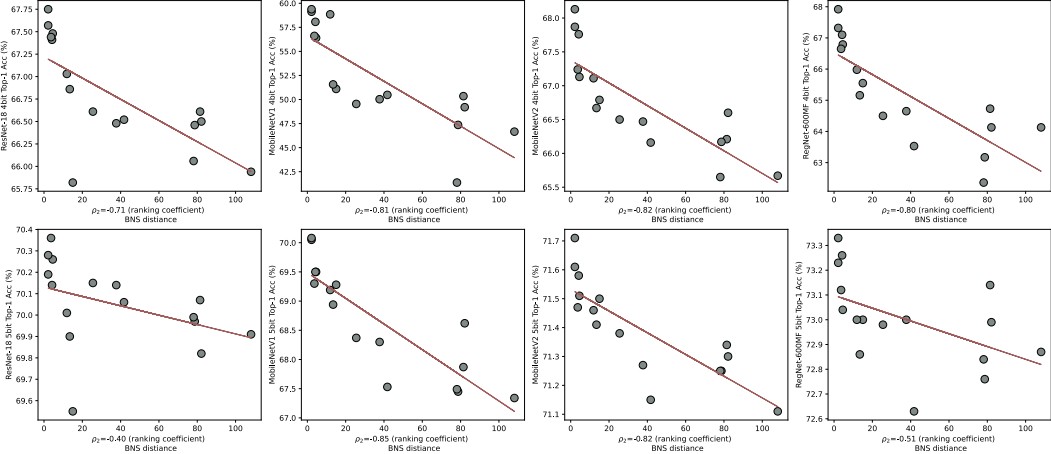

Figure 6: **The correlation of BNS distance and accuracy.** Pearson correlation coefficient ($\rho_2$), measures the fidelity of BNS distance for choosing an appropriate PD. The figure shows the $\rho_2$ for 4 architectures on 4-bit and 5-bit setting with ZeroP as baseline.

# B ADDITIONAL EXPERIMENT RESULTS

The following section is organized as follows. We will first present the extra experiments of Sec. 2 to further support our research flow, including the gains of PDs in 5-bit quantization settings in Sec. B.1.1 and the Pearson correlation coefficient of BNS and the accuracy in Sec. B.1.2. Then, we report the image classification performance comparisons with SOTA methods on CIFAR10/CIFAR100 in Sec. B.2. Finally, we present a comprehensive analysis of PD and ZeroP in Sec. B.3. We encourage the readers to refer to the sections for a comprehensive understanding of PD and ZeroP. Particularly, readers can directly hit into the **Analysis** section for the analysis experiment result of PD and ZeroP.

Table 5: **Image classification performance comparisons with SOTA methods on CIFAR10.** 'BW' means Bit-Width. This table report 3-bit, 4-bit, and 5-bit quantization settings. '*' denotes our implementation results based on the official code.

| Dataset | Model (FP32 Acc) | BW | GDFQ | Qimera | IntraQ | AIT | HAST | ZeroP$^{w/o}$ | ZeroP$^{w/}$ |
|---|---|---|---|---|---|---|---|---|---|
| CIFAR10 | ResNet-20 94.03 | 5w5a | 93.39* | 92.46 | 93.28 | 93.46 | - | 93.72 | **93.73** |
| | | 4w4a | 90.25 | 91.26 | 91.49 | 91.23 | 92.36 | 92.24 | **93.00** |
| | | 3w3a | 70.98* | 77.64 | 77.07 | 80.49 | **88.34** | 79.43 | 88.24 |
| CIFAR100 | ResNet-20 70.33 | 5w5a | 67.45* | 69.02 | 68.17 | 69.26 | - | **69.53** | 69.52 |
| | | 4w4a | 63.80 | 65.10 | 64.98 | 65.80 | 66.68 | 66.76 | **67.64** |
| | | 3w3a | 49.62* | 47.44 | 48.25 | 48.64 | 55.67 | 52.25 | **57.46** |

### B.1 Exploring the gains of Proxy Data

#### B.1.1 Additional results on Proxy Data

**Settings.** The experimental setting adheres to Sec. 4.1.1. We construct a simple ZeroP using only SD as a baseline and 16 different datasets as PD for the ZeroP baseline. For brevity, we use ZeroP$^{w/o}$ and ZeroP$^{w/}$ to indicate ZeroP using only SD and ZeroP using PD, respectively. We evaluate ZeroP$^{w/}$ on ResNet-18, MobileNetV1, MobileNetV2, and RegNet-600MF in 4-bit (4w/4a) and 5-bit (5w/5a) quantization settings. The result of the top-1 accuracy on the ImageNet-1K validation in 4-bit and 5-bit are reported in Tab. 1 and Tab. 4, repetively.

**Results.** As we can see from Tab. 4, we have almost the same consistent conclusion as the 4-bit case in Tab. 1. As the BNS distance increases, the performance gain decreases. Some datasets, *e.g.* CIFAR10, have larger BNS distances than Random Noise, yet they still yield better performance.

#### B.1.2 Selecting Proxy Data with BNS

**Settings.** The experimental setting is adhere to Sec. 4.1.2. We randomly selected 1024 images, $M = 1024$ in Equ. 5, from a specific PD and calculated the BNS distance using Equ. 5. We calculated the BNS distance for 16 PDs, and the results are listed in Tab. 1. In Sec. 4.1.2, we presented the Spearman's rank correlation coefficient ($\rho_1$) of BNS and accuracy in Fig. 4. Here we further present the Pearson correlation coefficient ($\rho_2$) of BNS and accuracy in Fig. 6.

**Results.** As we can see from Fig. 4 and Fig. 6, the BNS distance is highly related to the final performance. Also, $\rho_1$ and $\rho_2$ have a similar trend. For example, for ResNet-18 and RegNet-600MF in the 5-bit setting, both Spearman's rank correlation coefficient $\rho_1$ and Pearson correlation coefficient $\rho_2$ show relatively lower values. This result suggested that the indicator of PD selection can be further improved.

### B.2 Performance comparison

**Settings.** We report the comparison with SOTA methods on ImageNet-1K in Tab. 2. Here, we report the comparison result for CIFAR10/CIFAR100 in 3-bit, 4-bit, and 5-bit in Tab. 5 to support the superiority of ZeroP further. Since no OD SOTA methods, *e.g.* FDDA, BRECQ, provide results on CIFAR10/CIFAR10, we only compare with the pure-SD SOTA methods, such as HAST (Li et al., 2023), AIT (Choi et al., 2022), IntraQ (Zhong et al., 2022b), Qimera (Choi et al., 2021), and GDFQ (Xu et al., 2020).

**Results.** As we can see from Tab. 5, ZeroP, with SD and PD, outperforms almost all pure SD methods except for the HAST on CIFAR10 in the 3-bit case. For instance, ZeroP achieves performance increases of 1.79% for ResNet-20 in comparison with HAST on CIFAR100 for a 3-bit setting.

Table 6: **The generalization of ZeroP and the ablation study on PD.** Where 'RN' indicates 'Random Noise'. 'SD', 'PD', 'OD', and 'BW' share the same defination as before Tables. For each method, every 'SD' and 'PD' columns can be consider as showing the generalization of ZeroP. For each method, 'RN', 'SD', 'PD', and 'OD' can be viewed as the ablation on PD.

| Dataset | Model (FP32 Acc) | BW | GDFQ +SD | GDFQ +PD | Qimera +SD | Qimera +PD | IntraQ +SD | IntraQ +PD | ZeroP +SD | ZeroP +PD |
|---|---|---|---|---|---|---|---|---|---|---|
| CIFAR10 | ResNet-20 94.03 | 5w5a | 93.39* | 93.60(0.21↑) | 93.46 | **93.77**(0.31↑) | 93.28* | 93.35(0.07↑) | 93.72 | 93.73(0.01↑) |
| | | 4w4a | 90.25 | 92.53(2.28↑) | 91.26 | 92.81(1.55↑) | 91.49 | 91.84(0.35↑) | 92.24 | **93.00**(0.76↑) |
| | | 3w3a | 70.98* | 87.15(16.17↑) | 77.64* | 87.48(9.84↑) | 77.07 | 85.63(8.56↑) | 79.43 | **88.24**(8.81↑) |
| CIFAR100 | ResNet-20 70.33 | 5w5a | 67.45* | 68.04(0.59↑) | 69.02 | **69.56**(0.54↑) | 68.17* | 68.66(0.49↑) | 69.53 | 69.52(0.01↓) |
| | | 4w4a | 63.80 | 66.31(2.51↑) | 65.10 | 67.41(2.31↑) | 64.98 | 66.15(1.17↑) | 66.76 | **67.64**(0.88↑) |
| | | 3w3a | 49.62* | 55.40(5.78↑) | 47.44* | 56.32(8.88↑) | 48.25 | 53.68(5.43↑) | 52.26 | **57.46**(5.20↑) |
| ImageNet-1K | ResNet-18 71.47 | 5w5a | 68.24* | 69.27(1.03↑) | 69.29 | 70.12(0.83↑) | 69.94 | 70.24(0.30↑) | 69.92 | **70.28**(0.36↑) |
| | | 4w4a | 60.60 | 62.50(1.90↑) | 63.84 | 66.52(2.68↑) | 66.47 | 67.67(1.20↑) | 66.35 | **67.75**(1.40↑) |
| | ResNet-50 77.73 | 5w5a | 71.90* | 75.59(3.69↑) | 75.32 | 76.06(0.74↑) | 74.64* | 75.83(1.19↑) | 75.36 | **76.10**(0.74↑) |
| | | 4w4a | 56.04* | 65.98(9.94↑) | 66.25 | 68.62(2.37↑) | 56.88* | 70.88(14.00↑) | 64.50 | **72.17**(7.67↑) |
| | MobileNetV1 73.39 | 5w5a | 59.34* | 65.02(5.68↑) | 61.89* | 69.11(7.22↑) | 68.17 | 69.81(1.64↑) | 67.30 | **70.08**(2.78↑) |
| | | 4w4a | 30.79* | 38.41(7.62↑) | 38.66* | 53.00(14.34↑) | 51.36 | 57.21(5.85↑) | 43.31 | **59.38**(16.07↑) |
| | MobileNetV2 72.49 | 5w5a | 67.85* | 70.14(2.29↑) | 70.45 | 71.48(1.03↑) | 71.28 | 71.79(0.51↑) | 71.23 | **71.71**(0.48↑) |
| | | 4w4a | 59.56* | 64.63(5.07↑) | 61.72 | 65.72(4.00↑) | 65.10 | 67.73(2.63↑) | 63.70 | **68.13**(4.43↑) |

## B.3 ANALYSIS

### B.3.1 GENERALIZATION OF ZEROP

**Settings.** In this experiment, we demonstrate that the ZeroP pipeline can be easily generalized to other methods. The experiments are reported in Tab.6, and the settings are aligned with the PD ablation study described in Sec. 4.3. We focus on the 'SD' and 'PD' cases and highlight all the performance increases between the control experiments using a green upper row ↑ and a blue down row ↓ for performance increasing and decreasing, respectively.

**Results.** As shown in Tab. 6, it is worth noticing that, with the ZeroP pipeline, GDFQ, Qimera, and IntraQ obtain consistent accuracy improvements on CIFAR10, CIFAR100, and ImageNet-1K for all tested network architectures. In particular, IntraQ achieves a 14% improvement, Qimera achieves a 14.34% improvement, and GDFQ achieves a 9.94% improvement for ResNet-50, MobileNetV1, and ResNet-50 on ImageNet-1K in 4-bit quantization, respectively. These results demonstrate that the ZeroP pipeline can easily integrate into other ZSQ methods using SD and significantly improve performance.

Table 7: **The analysis of the ratio $\gamma$.** Baseline method is ZeroP with COCO as PD and ImageNet-1K as OD. The result of 5 different $\gamma$ values test on ResNet-18 and ResNet-50 are reported.

| BW | $\gamma$ | ResNet-18 | ResNet-50 |
|---|---|---|---|
| 5w5a | 0.0 | 69.92 | 75.36 |
| | 0.25 | **70.43** | 76.09 |
| | 0.50 | 70.28 | 76.10 |
| | 0.75 | 70.23 | 76.26 |
| | 1.0 | 70.28 | **76.60** |
| 4w4a | 0.0 | 66.35 | 64.50 |
| | 0.25 | 67.65 | 71.63 |
| | 0.50 | 67.75 | 72.17 |
| | 0.75 | 67.71 | 71.63 |
| | 1.0 | **68.26** | **72.83** |

Table 8: **The analysis of the ratio $\gamma$.** Baseline method is IntraQ with COCO as PD and ImageNet-1K as OD. The result of 5 different $\gamma$ values test on ResNet-18 and ResNet-50 are reported.

| BW | $\gamma$ | ResNet18 | ResNet50 |
|---|---|---|---|
| 5w5a | 0.0 | 69.94 | 74.64 |
| | 0.25 | 70.18 | 75.85 |
| | 0.50 | **70.24** | 75.83 |
| | 0.75 | 70.23 | 76.53 |
| | 1.0 | 70.10 | **75.90** |
| 4w4a | 0.0 | 66.47 | 56.88 |
| | 0.25 | 67.48 | 66.79 |
| | 0.50 | 67.67 | **70.88** |
| | 0.75 | **67.76** | 69.61 |
| | 1.0 | 67.71 | 69.86 |

### B.3.2 ARE SYNTHETIC DATA NEEDED?

**Settings.** In this experiment, we aim to explore the input data of ZeroP and how the ratio of SD and PD affects the final performance. As mentioned in Equ. 6, the input data of ZeroP is $\bar{x}$, and the

parameter $\gamma$ controls the ratio of SD and PD. We conduct experiments by using COCO as PD and set $\gamma$ to 0.0, 0.25, 0.50, 0.75, and 1.0, respectively. Here, $\gamma = 1.0$ means that only PD is used as the input data of ZeroP, $\gamma = 0.5$ means that SD and PD contributed equally to the input data, and $\gamma = 0.0$ means that only SD is used as the input of ZeroP. We report the results in 4-bit and 5-bit quantization settings.

**Results.** Our experimental results are presented in Tab. 7. We observe that the pure-SD ($\gamma = 0.0$) setting leads to worse performance across all cases. As we noted before, this may be due to the limitations of SD-based zero-shot quantization methods, which rely on a few single-angle loss terms to approximate the complex distribution of OD. On the other hand, the mixing case ($\gamma \in (0.0, 1.0)$) consistently outperforms the pure-SD case, which may be because PD captures rich properties of object representation. Another interesting observation is that, except for the ResNet-18 model in the 5-bit quantization setting, which achieves the highest accuracy when $\gamma = 0.25$, all other models achieve the best performance when $\gamma = 1.0$, indicating that using only PD leads to superior performance. We argue that this may be because the PD dataset (COCO) has similar properties to the OD dataset (ImageNet-1K). As shown in Tab. 8, the mixing case ($\gamma \in (0.0, 1.0)$) have the best performance in most case. All the results indicate that PD has the potential to handle data-lacking issues for data-free tasks. These results highlight the importance of using PD for data-free tasks and suggest that PD may be underestimated in the community.

### B.3.3 How the size of Proxy Data impact?

**Setting.** It is important to consider the resolution of the PD when using PD, as PD may not provide a similar scale object as the OD. However, in most CV tasks, the community has widely explored multi-scale settings. Also, as one may notice in Tab. 1, CIFAR10/CIFAR100 always performs worse than other candidate PDs or RN. Therefore, we investigate the impact of the resolution of the PD on ZeroP's performance. We use COCO as the PD for ImageNet-1K as OD

Table 9: **The analysis of the size of PD.** The baseline method is ZeroP with COCO as PD and ImageNet-1K as OD. The results of 5 different size are reported.

| BW | Size | ResNet-18 | MobileNetV1 | MobileNetV2 | RegNet-600MF |
|---|---|---|---|---|---|
| | 224×224 | 71.47 | 73.39 | 72.49 | 73.71 |
| 5w5a | 32×32 | 70.02 | 67.72 | 71.27 | 72.95 |
| | 64×64 | 70.16 | 68.59 | 71.32 | 73.03 |
| | 128×128 | 69.90 | 68.53 | 71.43 | 73.05 |
| | 160×160 | 70.03 | 69.09 | 71.20 | 73.12 |
| | 224×224 | **70.28** | **70.08** | **71.71** | **73.33** |
| 4w4a | 32×32 | 66.46 | 48.82 | 66.31 | 64.26 |
| | 64×64 | 66.56 | 51.88 | 66.65 | 65.16 |
| | 128×128 | 66.46 | 53.46 | 67.08 | 65.34 |
| | 160×160 | 66.61 | 52.46 | 66.93 | 65.12 |
| | 224×224 | **67.75** | **59.38** | **68.13** | **67.92** |

and randomly crop square image patches of different sizes using the $Resize(\cdot)$ function in torchvision (). Specifically, we crop the image patches with side lengths of 32, 64, 128, 160, and 224 and then resize them to 224×224 to combine them with SD as the final input data for ZeroP. We conduct experiments on ResNet-18, MobileNetV1, MobileNetV2, and RegNet-600MF for 4-bit and 5-bit settings. We report all results in Tab. 9.

**Results.** As shown in Tab. 9, the performance of the PD is roughly proportional to its size, indicating that the size similar to the OD's input data may lead to better performance. For instance, the best performance is achieved with the largest patch size of 224×224 for both 4-bit and 5-bit settings across all models. However, there is a considerable performance gap between the smallest patch size of 32×32 and the largest patch size of 224×224, *e.g.*, with a performance gap of 10.56% of MobileNetV1 in the 4-bit setting. Therefore, choosing a PD similar in features to the OD is essential, even though PDs are publicly available resources. In conclusion, the resolution of the PD should be carefully considered in ZeroP, and choosing a PD with similar features to the OD can lead to better performance.

### B.3.4 Does model architecture impact BNS-Based selection?

**Settings.** In this experiment, we investigate whether the BNS distance obtained by different model architectures affects the ranks of PDs over the same OD. We aim to determine if the BNS distance remains consistent across different models, which would simplify the process of selecting the appropriate PD. We calculate the BNS distance of ResNet-18, ResNet-50, MobileNetV1, MobileNetV2, and RegNet-600MF on 16 CV datasets. To assess the relative ordering of PDs, we plot the normal-

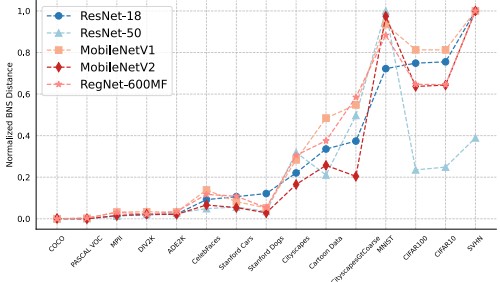
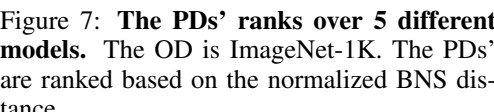
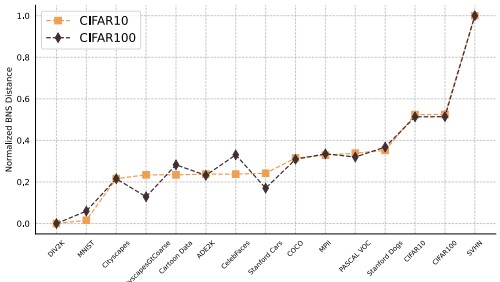

Figure 7: **The PDs' ranks over 5 different models.** The OD is ImageNet-1K. The PDs' are ranked based on the normalized BNS distance.

Figure 8: **The PDs' ranks of CIFAR10, CIFAR100 (OD).** The BNS distance calulate on ResNet-20.

ized BNS distance of different PDs in Fig. 7. We compute the normalized BNS distance $d'$ via: $d'_i = \frac{d_i - \min(\boldsymbol{d})}{\max(\boldsymbol{d}) - \min(\boldsymbol{d})}$, where $\boldsymbol{d} = [d_1, \ldots, d_n]$ represents the BNS distance of PDs.

**Results.** As shown in Fig. 7, the PDs' ranks of different model architectures exhibit a similar trend overall. Specifically, the PDs that perform better on the same OD tend to have a smaller BNS distance on different models, such as COCO, PASCAL VOC, MPII, and DIV2K. Conversely, the worse PDs for the same OD generally have larger BNS distance across all models, *e.g.*, MINIST, SVHN, Cartoon Data. It is worth noting that although the normalized BNS distance of CIFAR10/CIFAR100 on ResNet-50 is lower than that of other models, it is still larger than the top-tier PDs (*e.g.*, with normalized BNS distance ¡ 0.2), which may have a limited impact on the final PD selection. Based on these results, we conclude that the BNS distance is robust across different models, which facilitates the process of PD selection. We also reported the PDs' ranks of CIFAR10/CIFAR100 (as OD) in Fig. 8. As we can see, the PDs' ranks of CIFAR10/CIFAR100 are similar. Therefore, we use the same PD for CIFAR10 and CIFAR100 when they are used as OD in our experiments.

### B.3.5 CAN BNS SELECTION BE GENERALIZED TO OTHER METHODS?

**Settings.** In this experiment, we investigate whether the best candidate PDs selected by BNS for one OD can be generalized to other methods for the same OD, which could significantly reduce the cost of PD selection. We first filter 6 candidate PDs from three different distance groups based on their performance on 16 CV datasets, including CIFAR10, CIFAR100, Cityscapes, StanfordCars, and COCO, as shown in Tab. 1. We then apply the 6 selected PDs to three pure-SD ZSQ methods: GDFQ, Qimera, and IntraQ, using ImageNet-1K as the OD. For CIFAR10/CIFAR100 as OD, we use SVHN, CIFAR10/CIFAR100, ADE20K, and DIV2K as candidate PDs, and apply the same PDs to the same three pure-SD methods. We report the result that generalizes the BNS selection to IntraQ, Qimera, and GDFQ on ImageNet-1K in Tab. 10, Tab. 11, and Tab. 12, respectively. Besides, we also generalize the BNS selection of ZeroP to IntraQ, Qimera, and GDFQ on CIFAR10/CIFAR100. The gain of 5 PDs on CIFAR10/CIFAR100 (OD) based on ZeroP is shown in Tab. 16. The result that generalizes the BNS selection to IntraQ, Qimera, GDFQ on CIFAR10/CIFAR100 in Tab. 13, Tab. 14, and Tab. 15, respectively.

**Results.** As shown in Tab. 10, Tab. 11, and Tab. 12, we observe that the best candidate PD (COCO) for ZeroP also shows the best performance for Intra, Qimera, and GDFQ in the most case except for GDFQ of ResNet-18 in the 4-bit setting. Also, the large BNS distance PDs show inferior accuracy compared to COCO. Meanwhile, as the BNS distance increases, the selected PDs provide less performance gain for ZeroP. Next, when apply the BNS selection on CIFAR10/CIFAR100 as shown in Tab. 13, Tab. 14, and Tab. 15, ZeroP's BNS selection (DIV2K) does not always perform best on other pure-SD methods. However, DIV2K still shows promising approximate to the OD, CIFAR10/CIFAR100. Overall, those results on ImageNet-1K and CIFAR10/CIFAR100 align with the fact that PDs are approximations of the OD and that as long as the OD remains the same, the approximate relationship should remain the same when the method changes.

Table 10: **Generailze the PD selection of ZeroP to IntraQ with ImageNet-1K as OD.** The below 6 PDs are choosing according to the ZeroP's 16 dataset BNS distance.

| BW | ProxyData | BN | ResNet-18 | ResNet-50 | MobileNetV1 | MobileNetV2 |
|---|---|---|---|---|---|---|
| | FP32 Acc | 0 | 71.47 | 73.39 | 72.49 | 73.71 |
| | IntraQ | - | 69.94 | 74.64 | 68.17 | 71.28 |
| 5w5a | CIFAR10 | 82.14 | 69.70 | - | 67.65 | 71.06 |
| | CIFAR100 | 81.47 | 69.78 | - | 67.83 | 70.91 |
| | Random Noise | 78.09 | 69.98 | 73.78 | 67.70 | 71.24 |
| | Cityscapes | 25.53 | 70.17 | - | 68.99 | 71.70 |
| | StanfordCars | 13.45 | 69.96 | - | 68.94 | 71.70 |
| | COCO | **2.17** | **70.24** | **75.38** | **69.81** | **71.79** |
| | IntraQ | - | 66.47 | 56.88 | 51.36 | 65.10 |
| 4w4a | CIFAR10 | 82.14 | 64.85 | - | 51.93 | 64.97 |
| | CIFAR100 | 81.47 | 64.76 | - | 71.34 | 64.92 |
| | Random Noise | 78.09 | 65.56 | 59.16 | 49.54 | 65.72 |
| | Cityscapes | 25.53 | 66.99 | - | 51.80 | 66.54 |
| | StanfordCars | 13.45 | 67.12 | - | 53.16 | 66.72 |
| | COCO | **2.17** | **67.67** | **70.88** | **57.21** | **67.73** |

Table 11: **Generailze the PD selection of ZeroP to Qimera with ImageNet-1K as OD.** The below 6 PDs are choosing according to the ZeroP's 16 dataset BNS distance.

| BW | ProxyData | BN | ResNet-18 | ResNet-50 | MobileNetV1 | MobileNetV2 |
|---|---|---|---|---|---|---|
| | FP32 Acc | 0 | 71.47 | 73.39 | 72.49 | 73.71 |
| | Qimera | - | 69.29 | 75.32 | 61.89 | 70.45 |
| 5w5a | CIFAR10 | 82.14 | 69.32 | 74.62 | 66.12 | 70.56 |
| | CIFAR100 | 81.47 | 69.17 | 74.19 | 65.95 | 70.78 |
| | RandomNoise | 78.09 | 68.98 | 73.77 | 63.97 | 69.46 |
| | Cityscapes | 25.53 | 69.54 | 75.42 | 66.56 | 70.88 |
| | StanfordCars | 13.45 | 69.55 | 74.74 | 65.76 | 70.66 |
| | COCO(8) | **2.17** | **70.12** | **76.06** | **69.11** | **71.48** |
| | Qimera | - | 63.84 | 66.25 | 38.66 | 61.72 |
| 4w4a | CIFAR10 | 82.14 | 64.54 | 65.19 | 42.71 | 63.16 |
| | CIFAR100 | 81.47 | 63.74 | 65.83 | 42.99 | 62.32 |
| | RandomNoise | 78.09 | 63.19 | 61.71 | 41.58 | 59.39 |
| | Cityscapes | 25.53 | 63.89 | 65.04 | 44.18 | 63.11 |
| | StanfordCars | 13.45 | 64.53 | 64.85 | 43.80 | 63.25 |
| | COCO | **2.17** | **66.52** | **68.62** | **53.00** | **65.72** |

Table 12: **Generailze the PD selection of ZeroP to GDFQ with ImageNet-1K as OD.** The below 6 PDs are choosing according to the ZeroP's 16 dataset BNS distance.

| BW | ProxyData | BN | ResNet-18 | ResNet-50 | MobileNetV1 | MobileNetV2 |
|---|---|---|---|---|---|---|
| | FP32 Acc | 0 | 71.47 | 73.39 | 72.49 | 73.71 |
| | GDFQ | - | 68.24 | 71.90 | 59.34 | 67.85 |
| 5w5a | CIFAR10 | 82.14 | 68.49 | 70.65 | 61.09 | 69.34 |
| | CIFAR100 | 81.47 | 68.40 | 72.30 | 61.05 | 69.66 |
| | Random Noise | 78.09 | 68.37 | 72.86 | 59.54 | 68.22 |
| | Cityscapes | 25.53 | 69.21 | 74.18 | 61.32 | 69.14 |
| | StanfordCars | 13.45 | 68.76 | 73.73 | 59.21 | 69.16 |
| | COCO | **2.17** | **69.27** | **75.59** | **65.02** | **70.14** |
| | GDFQ | - | 60.60 | 56.04 | 30.79 | 59.56 |
| 4w4a | CIFAR10 | 82.14 | 61.54 | 58.88 | 25.41 | 60.64 |
| | CIFAR100 | 81.47 | 61.89 | 57.50 | 26.94 | 61.50 |
| | Random Noise | 78.09 | 60.51 | 50.85 | 30.05 | 59.55 |
| | Cityscapes | 25.53 | **62.96** | 61.83 | 27.76 | 62.26 |
| | StanfordCars | 13.45 | 59.26 | 58.13 | 23.55 | 61.90 |
| | COCO | **2.17** | 62.50 | **65.98** | **38.41** | **64.63** |

Table 13: **Generailze the PD selection of ZeroP to IntraQ ResNet-20 with CIFAR10 and CIFAR100 as OD.** The below 5 PDs are the same as the ZeroP's 5 dataset BNS distance.

| BW | ProxyData | BN | 3w3a | 4w4a | 5w5a |
|---|---|---|---|---|---|
| | IntraQ | - | 77.07 | 91.49 | 93.28 |
| CIFAR10 | Random Noise | 960.10 | 75.88 | 90.76 | 93.33 |
| (94.03) | SVHN | 14.57 | 75.26 | 91.28 | 93.22 |
| | CIFAR100 | 81.47 | 83.89 | 91.63 | **93.43** |
| | ADE2K | 5.36 | **85.63** | **91.84** | 93.35 |
| | DIV2K | **2.49** | 85.06 | 91.82 | 93.06 |
| | IntraQ | - | 48.25 | 64.98 | 68.17 |
| CIFAR100 | Random Noise | 2218.47 | 48.45 | 64.74 | 67.92 |
| (70.33) | SVHN | 69.02 | 48.52 | 65.17 | 68.06 |
| | CIFAR10 | 41.05 | 52.80 | 65.74 | 68.13 |
| | ADE2K | 24.88 | 53.68 | **66.15** | 68.66 |
| | DIV2K | **11.56** | **54.27** | 66.14 | **68.99** |

Table 14: **Generailze the PD selection of ZeroP to Qimera ResNet-20 with CIFAR10 and CIFAR100 as OD.** The below 5 PDs are the same as the ZeroP's 5 dataset BNS distance.

| BW | ProxyData | BN | 3w3a | 4w4Aa | 5w5a |
|---|---|---|---|---|---|
| | Qimera | - | 77.64 | 91.26 | 93.46 |
| CIFAR10 | Random Noise | 960.10 | 73.43 | 88.70 | 92.50 |
| (94.03) | SVHN | 14.57 | 79.74 | 91.46 | 93.13 |
| | CIFAR100 | 81.47 | 87.76 | 92.88 | 93.72 |
| | ADE2K | 5.36 | 87.48 | 92.81 | **93.77** |
| | DIV2K | **2.49** | **88.00** | **92.98** | 93.73 |
| | Qimera | - | 47.44 | 65.10 | 69.02 |
| CIFAR100 | Random Noise | 2218.47 | 43.84 | 61.62 | 66.41 |
| (70.33) | SVHN | 69.02 | 51.01 | 66.23 | 69.10 |
| | CIFAR10 | 41.05 | 57.19 | **67.59** | **69.79** |
| | ADE2K | 24.88 | 56.32 | 67.41 | 69.56 |
| | DIV2K | **11.56** | **58.03** | 67.58 | 69.58 |

Table 15: **Generailze the PD selection of ZeroP to GDFQ ResNet-20 with CIFAR10 and CIFAR100 as OD.** The below 5 PDs are the same as the ZeroP's 5 dataset BNS distance.

| BW | ProxyData | BN | 3w3a | 4w4a | 5w5a |
|---|---|---|---|---|---|
| | GDFQ | - | 70.98 | 90.05 | 93.39 |
| CIFAR10 | Random Noise | 960.10 | 67.34 | 88.04 | 92.13 |
| (94.03) | SVHN | 14.57 | 76.26 | 90.33 | 93.06 |
| | CIFAR100 | 81.47 | 87.32 | 92.76 | 93.64 |
| | ADE2K | 5.36 | 87.15 | 92.53 | 93.60 |
| | DIV2K | **2.49** | **87.87** | **92.82** | **93.68** |
| | GDFQ | - | 49.62 | 63.80 | 67.45 |
| CIFAR100 | Random Noise | 2218.47 | 36.19 | 57.61 | 65.55 |
| (70.33) | SVHN | 69.02 | 50.31 | 64.63 | 67.53 |
| | CIFAR10 | 41.05 | **57.68** | 65.95 | 67.67 |
| | ADE2K | 24.88 | 55.40 | 66.31 | **68.04** |
| | DIV2K | **11.56** | 56.71 | **66.34** | 67.24 |

Table 16: **The gain of 5 PDs on CIFAR10/CIFAR100 (OD) based on ZeroP.** We report 5 different BNS distance PD's performances and generalize them to other methods.

| BW | ProxyData | BN | 3w3a | 4w4Aa | 5w5a |
|---|---|---|---|---|---|
| | ZeroP$^{w/o}$ | - | 79.43 | 92.24 | 93.72 |
| CIFAR10 | Random Noise | 960.10 | 78.23 | 91.40 | 93.49 |
| (94.03) | SVHN | 14.57 | 79.74 | 91.89 | 93.57 |
| | CIFAR100 | 81.47 | 87.84 | 92.82 | **93.86** |
| | ADE2K | 5.36 | 88.04 | 92.86 | 93.74 |
| | DIV2K | **2.49** | **88.24** | **93.00** | 93.73 |
| | ZeroP$^{w/o}$ | - | 52.25 | 66.76 | 69.53 |
| CIFAR100 | Random Noise | 2218.47 | 52.93 | 65.51 | 68.75 |
| (70.33) | SVHN | 69.02 | 55.12 | 66.61 | 69.10 |
| | CIFAR10 | 41.05 | **57.67** | 67.25 | 69.19 |
| | ADE2K | 24.88 | 56.32 | 67.24 | 69.17 |
| | DIV2K | **11.56** | 57.46 | **67.64** | **69.52** |