# OpenReview forum: "ZeroP: Zero-Shot Quantization via Proxy Data"
_ICLR.cc/2024/Conference — Submitted to ICLR 2024_

### Official Review · Reviewer_FpNc · 2023-10-13

**Soundness:** 3 good
**Presentation:** 3 good
**Contribution:** 2 fair
**Rating:** 5
**Confidence:** 5

**Summary:**

The paper proposes a novel ZSQ framework, leveraging publicly available data instead of  synthetic data.
It offers a promising solution for achieving high-performance low-bit networks without relying on original training data.

**Strengths:**

1. Leveraging open-world or public dataset for ZSQ is interesting and reasonable.
2. Comparing to ZSQ relying on synthetic data, ZeroP is more efficient and easier to implement.

**Weaknesses:**

1. The core contribution is somewhat limited. In my opinion, ZSQ is just a sub-area of data-free KD, and any data-free KD methods can be extend to ZSQ. [1][2] are two data-free KD methods that utilize proxy data for distillation. They can also be applied on ZSQ.
2. There is lack of more details. e.g. image number, of the proxy datasets. And there is not ablation on the numbers of proxy data.

## ref
[1] Sampling to Distill: Knowledge Transfer from Open-World Data, 2307.16601
[2] Learning Student Networks in the Wild, CVPR 2021

**Questions:**

Please refer to Weaknesses.

---

### Official Review · Reviewer_MuxB · 2023-10-22

**Soundness:** 2 fair
**Presentation:** 1 poor
**Contribution:** 2 fair
**Rating:** 3
**Confidence:** 4

**Summary:**

This paper leverages publicly available data, termed as Proxy Data (PD), as a substitute for original data (OD). The paper addresses the limitations of existing ZSQ methods that rely solely on synthetic data (SD) by introducing a method to select optimal PD based on batch-normalization statistics. The ZeroP framework is applied to existing pure-SD methods, resulting in significant improvements in accuracy. Specifically, ZeroP outperforms state-of-the-art pure-SD methods by 3.9% in a 4-bit setting for ResNet-50 on ImageNet-1K. The paper also introduces a simple and effective method for guiding PD selection, thereby offering a promising solution for achieving high-performance low-bit networks without relying on original data.

**Strengths:**

1. The paper introduces a new approach to ZSQ by incorporating publicly available Proxy Data, filling a gap in the existing literature. A comprehensive methodology is provided, including a PD selection method based on batch-normalization statistics, which adds to its credibility.
2. ZeroP shows significant improvements in accuracy over existing methods in low-bit settings.

**Weaknesses:**

1. While the paper discusses improvements in accuracy, it does not provide sufficient information on the scalability of the proposed method, especially when dealing with larger datasets or more complex models.
2. Lack of performance in low-bit settings, such as 2-bit and 1-bit. I wonder whether the methods used PD can have a competitive performance over the previous quantization/binarization methods.
3. It is better to provide the preliminary knowledge of the proxy data, and how previous work uses the proxy data for the quantization.

**Questions:**

1. How well does the proposed ZeroP framework generalize to other types of neural networks or tasks beyond image classification?
2. As for the computational overhead, could you elaborate on the computational cost involved in the PD selection process, and how the computational overhead of the selection of PD compared with the computations in the training process?

---

### Official Review · Reviewer_7Mby · 2023-10-31

**Soundness:** 3 good
**Presentation:** 3 good
**Contribution:** 2 fair
**Rating:** 5
**Confidence:** 4

**Summary:**

The paper introduces a new quantization-aware finetuning method for visual recognition that does not rely on the original training data (OD). The proposed method, ZeroP, instead leverages realistic proxy data (PD) in addition to the conventional synthetic data (SD) to further finetune the model for quantization. Here, incorporating PD based on the batchnorm statistic (BNS) is the key contribution of the paper. Experimental results show that ZeroP outperforms SD-only approaches and performs on par with OD-based works.

**Strengths:**

(S1) [Motivation] Going beyond synthetic data for zero-shot quantization is interesting. The reviewer agrees with the author that it is not necessary to rely solely on synthetic data, especially when relevant  information of the target task is available.

(S2) [Performance] The proposed method demonstrates superior performance.

(S3) [Ablation] Ablations show that PD could be a plug-in solution that helps improve the performance of SD-only methods in general.

(S4) [Writing] The paper is easy to follow.

**Weaknesses:**

(W1) The current method to select the optimal PD dataset is straightforward, i.e. ranking the PDs by the gap of the BNS. The technical contribution is weak.

(W2) Relying on BNS also limits the versatility of ZeroP (as also indicated in the Limitation section)

(W3) If I understood correctly, the key challenge here is to search for PDs that mimic the distribution of the OD. In this case, using only BNS may not be necessary. Depending on the target task, there may be more information we could make use of, e.g. the class names of the target task. (If the finetuning involves a classification loss, this information may already be available.) With such information, instead of searching for a specific PD dataset, we could search for relevant samples via a text-based search engine, e.g. CLIP.

Overall, the reviewer likes the idea of incorporating PD for zero-shot quantization, and also appreciates the superior performance of ZeroP. The reviewer has concerns about the technical contributions and the potential impacts of the paper. Therefore, the reviewer rates the paper as marginally below the acceptance threshold.

**Questions:**

N.A.

**Details Of Ethics Concerns:**

N.A.

---

### Official Review · Reviewer_AkjD · 2023-11-01

**Soundness:** 3 good
**Presentation:** 3 good
**Contribution:** 2 fair
**Rating:** 6
**Confidence:** 3

**Summary:**

The paper presents ZeroP, a novel approach for the Zero-Shot Quantization (ZSQ) task. The approach aims to investigate the potential gain of Proxy Data (PD) across 16 commonly used CV datasets. In addition, the paper introduces the BNS distance as a simple yet effective metric for selecting suitable PD for a specific task.

**Strengths:**

- The paper introduces the BNS distance metric which provides a simple yet effective means to select suitable Proxy Data for a given task.
- The paper conducts thorough experiments showing that ZeroP outperforms existing pure-SD methods by a significant margin across diverse datasets.
- The work is relevant given the need for efficient methods in the ZSQ space without relying on original data.

**Weaknesses:**

- The approach, while novel in certain aspects, leans heavily on established methodologies such as pure-SD. The introduction and utilization of Proxy Data, although effective, do not drastically deviate from methods previously explored in the domain of data-free tasks.
- The paper mainly focuses on 4-bit and 5-bit quantization, leaving questions about the performance and relevance of other bit quantizations.

**Questions:**

- The focus on 4-bit and 5-bit quantizations was evident, but it raises the question: what about other bit depths? Were experiments conducted with other bit quantizations, and if so, what were the results? Elaborating on this could provide a broader understanding of the system's applicability.

---

### Official Review · Reviewer_qH2g · 2023-11-02

**Soundness:** 4 excellent
**Presentation:** 4 excellent
**Contribution:** 2 fair
**Rating:** 3
**Confidence:** 5

**Summary:**

A simple but intuitive method that uses proxy data for ZSQ. To find the most suitable proxy data, a BNS-based distance is used where a small BNS distance indicates a higher relation between proxy data and original data.

**Strengths:**

This method is simple but effective. It does provide a SOTA performance. Comprehensive and impressive experiment results.
This paper is valuable.

**Weaknesses:**

The novelty of this paper appears constrained, particularly when considered with an earlier work that seemingly shares a similar idea.

[1] "Is In-Domain Data Really Needed? A Pilot Study on Cross-Domain Calibration for Network Quantization," CVPR2021Workshop.

Note that [1] is an accepted paper, not a preprint paper. However, I can't find any reference to [1] within this manuscript. While it's not feasible to reference every related work, the conceptual overlap with [1] is pronounced. [1] primarily targets PTQ, but it does not involve real data and can be regarded as a ZSQ method.  And the only different point is the select metric.

I think this paper is valuable. However, more experiments for comparison are needed.

**Questions:**

See weaknesses

---

### Meta-Review · Area_Chair_kCwZ · 2023-12-05

**Metareview:**

Four experts reviewed the paper. All but one review recommended to not accept the paper. The only positive reviewer still raised several major concerns. There was no rebuttal.

**Justification For Why Not Higher Score:**

All but one review recommended to not accept the paper. The only positive reviewer still raised several major concerns. There was no rebuttal.

**Justification For Why Not Lower Score:**

No lower score available.

---

### Decision · Program_Chairs · 2024-01-16

Reject